# Multivariate Time Series Forecasting By Graph Attention Networks With Theoretical Guarantees

## Abstract

Multivariate time series forecasting (MTSF) aims to predict future values of multiple variables based on past values of multivariate time series, and has been applied in fields including traffic flow prediction, stock price forecasting, and anomaly detection. Capturing the inter-dependencies among variables poses one significant challenge to MTSF. Several methods that model the correlations between variables with an aim to improve the test prediction accuracy have been considered in recent works, however, none of them have theoretical guarantees. In this paper, we developed a new norm-bounded graph attention network (GAT) for MTSF by upper-bounding the Frobenius norm of weights in each layer of the GAT model to achieve optimal performance. Under optimal parameters, we theoretically show that our model can achieve a generalization error bound which is expressed as products of Frobenius norm of weight in each layer and the numbers of neighbors and attention heads, while the latter is represented as polynomial terms with the degree as the number of layers. Empirically, we investigate the impact of different components of GAT models on the performance of MTSF. Our experiment also verifies our theoretical findings. Empirically, we also observe that the generalization performance of our method is dependent on the number of attention heads, the number of neighbors, the scales (norms) of the weight matrices, the scale of the input features, and the number of layers. Our method provides novel perspectives for improving the generation performance for MTSF, and our theoretical guarantees give substantial implications for designing attention-based methods for MTSF.

## 1 Introduction and Backgrounds

Substantial time series data generated in the real world make multivariate time series forcasting (MTSF) a crucial topic in various scenarios, such as traffic forecasting, sensor signal anomaly detection in the Internet of things, demand and supply prediction in the supply chain management, and stock market price prediction in financial investment (Cao et al., 2020). Traditional methods simply deploy time series models, e.g., auto-regressive (AR) (Mills & Mills, 1990), auto-regressive integrated moving average (ARIMA) (Box et al., 2015) and vector auto-regression (VAR)(Box et al., 2015; Hamilton, 2020; Lütkepohl, 2005) for forecasting. Specifically, ARIMA, though one of the classic forecasting methods in univariate situations, fails to accommodate multivariate issues due to high computational complexity. VAR, as an extension of AR model in multivariate situations, is widely used in MTSF tasks due to its simplicity, however, it cannot handle the nonlinear relationships among variables, leading to reduced forecasting accuracy. In addition to traditional statistical methods, deep learning methods have been applied in MTSF problems and demonstrated potentials to solve these problems (Tokgöz & Ünal, 2018).

The long short term memory (LSTM) (Graves, 2012), gated recurrent units (GRU) (Cho et al., 2014), gated linear units (GLU) (Dauphin et al., 2017), temporal convolution networks (TCN) (Bai et al., 2018), state frequency memory (SFM) network (Zhang et al., 2017) have found success in practical time series tasks. However, another important issue in time series data, complex inter-dependency (i.e., the correlations among multiple correlated time series), is still unaddressed in these methods, restricting forecasting accuracy (Bai et al., 2020; Cao et al., 2020). For example, in the traffic forecasting task, adjacent roads naturally interplay with each other. Another example is stock price prediction, in which it is easier to predict a stock price based on the historical information of the stocks in similar categories, while information on stocks from other sectors can be relatively useless.

Graph is a special form of data that describes the relationships between different entities. Recently, graph neural networks (GNNs)(Scarselli et al., 2008) have achieved great success in handling graph data with development in permutation-invariance, local connectivity, and compositionality. In general,

GNNs assume that the state of a node is influenced by the states of its neighbors. By disseminating information through structures, GNNs allow each node in a graph to be aware of its neighborhood context. MTSF can be viewed naturally from a graph perspective. Variables from multivariate time series can be considered as nodes in a graph where they are interlinked each other through hidden dependency relationships. It follows that modeling multivariate time series data using GNNs can be a promising way to preserve their temporal trajectory while exploiting the inter-dependency among time series. In the meantime, due to the popularity of convolutional neural networks (CNNs), considerable studies attempt to generalize convolutions to graph-structured data, leading to the creation of graph convolutional networks (GCNs) (Duvenaud et al., 2015; Atwood & Towsley, 2016; Monti et al., 2018; Niepert et al., 2016; Kipf & Welling, 2017). GCNs model a node's feature representation by aggregating the representations of its one-step neighbors. Many studies have shown that GNN- and GCN-based methods outperform prior methods in time series forecasting tasks (Yu et al., 2017; Wu et al., 2019; Chen et al., 2020).

The graph attention network (GAT) (Veličković et al., 2017), one of the most popular GNN architectures, is considered a state-of-the-art neural architecture to process graph-structured data. Building on the aggregating approach of GCNs, in GATs, every node computes the importance of its neighboring nodes, and then utilizes the importance as weights to update its representations of the features during the aggregation. Compared to the well-known GCNs, GATs have demonstrated equivalent, if not improved, performance across well-established benchmarks of node multiclass classification. Within the GAT framework, Guo et al. (2019); Deng & Hooi (2021) use GAT-based models to adaptively adjust the correlations among multiple time series, showing better performance in accuracy over GNNs and GCNs.

The numeric and experimental successes of GATs for MTSF notwithstanding, theoretical understandings of the underlying mechanisms of GATs for MTSF are still limited: none of them has theoretical guarantees with respect to generalization error bounds, the most commonly used method to theoretically evaluate the prediction model.

The generalization error bound provides a standard approach to evaluate neural networks as it characterizes the predictive performance of a class of learning models for unseen data (Golowich et al., 2018). Therefore, understanding the generalization error bound of GATs for MTSF will shed light on the relationship between the architecture of the GATs and their generalization performance for MTSF, advancing understandings of underlying mechanisms.

Studies show that deriving generalization error bound for neural network classes requires constraints on the size of weights. Bartlett (1998) first gave a generalization error bound for neural networks by bounding the size of the cover of neural network function classes, suggesting that the bound depends on the number of training samples and the size of weights, rather than the number of weights. In the following studies, the empirical Rademacher complexity (ERC) was shown as an essential component of generalization error bound for neural network classes. Bartlett & Mendelson (2002) introduced the generalization error bounds using the Rademacher complexity of the function classes that include neural networks with constraints on the magnitudes of weights for binary classification. Bartlett et al. (2017) then presented a margin-based generalization error bound using the Rademacher complexity of neural network function classes with the spectral norms of weight matrices being controlled for multiclass classification. Neyshabur et al. (2015) used the Rademacher complexity bounds showing that the generalization error of deep neural networks can be upper bounded by a bound in terms of the Frobenius norms of weights. Golowich et al. (2018) further demonstrated that the generalization error bound for deep neural network classes with bounded Frobenius norm of weights can be independent of the number of layers and the width of each layer if employing proper techniques. These methods are also extended to graph-based neural networks. Garg et al. (2020) derived the generalization error bounds for GNNs using Rademacher complexity for binary classification. Lv (2021) provided the generalization error bound for GCNs via Rademacher complexity for binary classification.

**Contributions.** In this study, to capture the inter-dependencies among variables of MTSF, we develop a GAT-based method for MTSF; to secure the generalization error bound, we require the norm of weight matrix in our model to be bounded; to evaluate the performance of our method, we compare our method with two SOTA methods and show our method outperforms over these prior methods. We also provide the theoretical generalization error bound for our method, aiming to develop models with a desired generalization error for MTSF.

Specifically, we derive the generalization error bounds of two-layer GAT models for multi-step MTSF task. We also extend our generalization error bounds to deep GAT models with more than two layers. Generalization error bounds derived in this study are based on the bound of ERC of GAT models with the weight matrix norm being controlled. This approach is characterized by controlling Frobenius norm of the hidden layer weight matrix, a common method to derive the norm-based generalization error bounds for DNNs, CNNs, and GNNs. In particular, we show that ERC derived for GAT models for MTSF has a polynomial dependence on the number of neighbors considered

in attention representation and the number of attention heads being used. The aforementioned ERC is also dependent on the product of norms of weight matrices of each layer, the $L_2$-norm of the input feature vector, and the Lipschitz constant of loss and activation functions. To further understand the effectiveness of GATs for MTSF, in addition to theoretical analysis of the relationships between components of the GAT model and this bound, we also investigate the influence of different components of GATs models on the performance of MTSF using experiments with complex stock price data. Our experimental results are consistent with theoretical findings. To our best knowledge, we develop the first GAT-based method for MTSF with theoretical guarantees.

## 2 PRELIMINARIES

### 2.1 PROBLEM FORMULATION

In this paper, we focus on the task of MTSF, considering a multivariate situation that contains $N$ correlated univariate time series represented as $\{\mathfrak{X}_1, \ldots, \mathfrak{X}_N\}$, where we use $\mathfrak{X}_i = \{\mathbf{x}_{i,1}, \mathbf{x}_{i,2}, \ldots, \mathbf{x}_{i,t}, \ldots\}$ to denote a sequence of time series $i$ from time step 1 to infinity. Based on a sequence of historical $T$ time steps of values prior to current time $t$, our goal is to predict the multi-step-away value of $\{\mathbf{y}_1, \ldots, \mathbf{y}_N\}$ using an appropriate prediction model $f$, where each $\mathbf{y}_i = \{\mathbf{y}_{i,t+1}, \ldots, \mathbf{y}_{i,t+C}\}$ has values from $C$ timestamps. In addition, the historical inputs can be representative of multiple aspects if complemented with auxiliary features, so our problem can be characterized as $\{\mathbf{y}_1, \ldots, \mathbf{y}_N\} = f(\{\mathbf{x}_{1,t}, \ldots, \mathbf{x}_{1,t-T+1}\}, \ldots, \{\mathbf{x}_{N,t}, \ldots, \mathbf{x}_{N,t-T+1}\})$. To accurately capture the inter-dependency, the problem is formulated on the graph structure as introduced below.

### 2.2 THE GRAPH STRUCTURE

Now we consider an undirected graph $\mathcal{G} = (\mathcal{N}, \mathcal{E})$. $\mathcal{N} = (n_1, \ldots, n_N)$, $|\mathcal{N}| = N$, is a set of node labels representing the sources of $N$ time series. $\mathcal{E} \subset \mathcal{N} \times \mathcal{N}$ is the set of edges representing the connection between series. We let $\mathbf{x}_i \in \mathcal{X}$, $i \in [N]$ be a random variable representing the input feature vector of node $n_i$ for time series $i$. For node $i$, its random input feature $\mathbf{x}_i \in \mathcal{X} \subset \mathbb{R}^D$ is a multi-dimensional vector, which contains all the historical values from $T$ time steps, in other words, we let $\mathbf{x}_i = (\mathbf{x}_{i,t}, \ldots, \mathbf{x}_{i,t-T+1})$ be the concatenation of $T$ time steps; its true label $\mathbf{y}_i \in \mathcal{Y} \subset \mathbb{R}^C$ is the vector for the C-step-away values. During the learning period, some nodes, which will be treated as the training set, know the C-step-away true values $\mathbf{y}$. We denote the set of indices of those nodes as $\mathcal{M} \subset \mathcal{N}$ such that $M = |\mathcal{M}| < N$. And for each node in the $\mathcal{M}$, we order them based on their node labels $n_i$ and re-index them based on their order number, $j \in [M]$. Here the random input and labels are $S = \{(\mathbf{x}_1, \mathbf{y}_1), (\mathbf{x}_2, \mathbf{y}_2), \ldots, (\mathbf{x}_M, \mathbf{y}_M)\}$ over $\mathcal{M}$. In the following paragraphs, we will introduce the GATs for our problem.

### 2.3 GATS MODEL

We consider the GATs defined by Veličković et al. (2017), given the random feature matrix $\mathbf{X} = [\mathbf{x}_1, \mathbf{x}_2, \ldots, \mathbf{x}_N]$, a $L$-layers GAT model $f$, and the final output $\mathbf{Z}_{(L)}$,

$$\mathbf{Z}_{(L)} = \left[f(\mathbf{x}_1)^\top, \ldots, f(\mathbf{x}_M)^\top\right]^\top$$
$$= \mathbf{P}_{(L)} \oplus_{k=1}^K \sigma\left(\mathbf{P}_{(L-1)} \cdots \oplus_{k=1}^K \sigma\left(\mathbf{P}_{(2)} \oplus_{k=1}^K \sigma\left(\mathbf{P}_{(1,k)}\mathbf{X}\mathbf{W}_{(1,k)}\right)\mathbf{W}_{(2)}\right) \cdots \mathbf{W}_{(L-1)}\right)\mathbf{W}_{(L)}.$$

In a GAT with more than two layers, the output of the hidden layer $l$ is

$$\mathbf{Z}_{(l)} = \oplus_{k=1}^K \sigma(\mathbf{P}_{(l,k)}\mathbf{Z}_{(l-1)}\mathbf{W}_{(l,k)}), \tag{1}$$

Here we use subscript $(l)$ or $(l-1)$ to indicate which layer the variable belongs to. We have $l \in \{1, 2, \ldots, L-1\}$, and $\mathbf{Z}_0 = \mathbf{X}$.

Here, $\mathbf{W}_{(l,k)} \in \mathbb{R}^{D_{l-1} \times D_l}$ is an $l-1$-to-$l$ weight matrix for a hidden layer with $D_l$ feature maps. Here $\sigma$ is the activation function.

And we let $\oplus$ denote the concatenation for the *attention heads*. This definition is specific to the GATs, with its detailed description found in Veličković et al. (2017). And we have total $K$ such matrices in layer $l$ with each matrix $\mathbf{W}_{(l,k)}$ corresponding to one attention head. $\mathbf{P}_{(1,k)}$, $\mathbf{P}_{(2)}$ and $\mathbf{P}_{(L-1)}$ are the *attention matrix* introduced by us, and further function as an operator to incorporate the attention. We will justify its equivalence to the original GAT models (Veličković et al., 2017) in later paragraphs.

Even though our analysis covers GATs with more than two layers, we will give a focus on two-layers GATs model, which is also implemented by Veličković et al. (2017), with the following simple form:

$$\mathbf{Z}_{(2)} = \left[ f(\mathbf{x}_1)^\top, \ldots, f(\mathbf{x}_M)^\top \right]^\top = \mathbf{P}_{(2)} \oplus_{k=1}^K \sigma \left( \mathbf{P}_{(1,k)} \mathbf{X} \mathbf{W}_{(1,k)} \right) \mathbf{W}_{(2)}. \tag{2}$$

The first layer consists of $K$ attention heads computing $D_1$ features each (a total of $K \times D_1$ features), followed by an activation function $\sigma$. Here, $\mathbf{W}_{(1,k)} \in \mathbb{R}^{D \times D_1}$ is an input-to-hidden weight matrix for a hidden layer with $D_1$ feature maps, and we have $K$ such matrices.

The second layer is used for prediction: a single attention head that predicts the $\mathbf{y}$. For C-step-away forecasting, we have $D_L = C$. The $\mathbf{W}_{(2)} \in \mathbb{R}^{KD_1 \times C}$ is a hidden-to-output weight matrix.

**Attention Model**    We now give more explanation about the *attention* introduced in the GAT model. In section 2 of the original GAT paper, Veličković et al. (2017) mentioned the *learnable linear transformation* to transform the input features into higher-level features with sufficient expressive power. In their process, they first apply a shared linear transformation, parameterized by the weight matrix, $\mathbf{W} \in \mathbb{R}^{D \times D_1}$, to every node. Then they perform *self-attention* on the node: a shared attentional mechanism $a : \mathbb{R}^{D_1} \times \mathbb{R}^{D_1} \to \mathbb{R}$ computes attention coefficients $e_{ij} = a(\mathbf{x}_i^\top \mathbf{W}, \mathbf{x}_j^\top \mathbf{W})$, to indicate the importance of node $j$'s features to node $i$. They also inject the graph structure into the mechanism by performing masked attention: they compute $e_{ij}$ for nodes $j \in \mathcal{N}(i)$, the neighbors of node $i$, which might include node $i$ itself. Then they normalize the coefficients to make them easily comparable across different nodes using the softmax function to obtain $p^{i,j}$

$$p^{i,j} = \phi([e_{i,1}, e_{i,2}, \ldots])^j = \frac{\exp e_{i,j}}{\sum_{k \in \mathcal{N}(i)} \exp e_{i,k}}.$$

Then the output features from the first layer for each node will be: $\sigma(\sum_{j \in \mathcal{N}(i)} p^{i,j} \mathbf{x}_j^\top \mathbf{W})$. They also propose the $K$-head attention. The $K$ independent attention mechanisms execute the aforementioned transformation, and then their output are concatenated, resulting in the following feature representation: $\oplus_{k=1}^K \sigma(\sum_{j \in \mathcal{N}(i)} p_{(k)}^{i,j} \mathbf{x}_j^\top \mathbf{W}_{(k)})$.

To make the above process more integrated, here in the GAT models, we introduce the attention matrix $\mathbf{P}$ that contains the normalized attention coefficients used to compute a linear combination of the neighborhood features, yielding the new feature representation for every node. This matrix contains individual node's importance weights with every other node in its neighborhood.

Let $\mathbf{P}_{(l)} \in \mathbb{R}^{N \times N}$, $l \in [L-1]$, is the matrix of the graph attention matrix defined by the attention coefficients [1]. We use $N_e \leq N$ as the fixed number of neighbors for each node. And let $\mathcal{N}(n)$ be the set of neighbors for node $n$.

$$\mathbf{P}_{(l)} = \begin{bmatrix} -\mathbf{p}_{(l)}^1- \\ -\mathbf{p}_{(l)}^2- \\ \vdots \\ -\mathbf{p}_{(l)}^N- \end{bmatrix} = \begin{bmatrix} \underbrace{0 \quad p_{(l)}^{1,i} \quad p_{(l)}^{1,k} \quad \cdots \quad p_{(l)}^{1,m}}_{\{i,k,m\}=\mathcal{N}(1)} \\ \vdots \\ \underbrace{p_{(l)}^{N,s} \quad p_{(l)}^{N,q} \quad \cdots \quad p_{(l)}^{N,j} \quad 0}_{\{s,q,j\}=\mathcal{N}(N)} \end{bmatrix},$$

$p_{(l)}^{n,e} \in [0,1]$ is the coefficients of node $n$ attributed from node $e$. Each row sum of the $\mathbf{P}_{(l)}$ is equal to 1, which is $\sum_{e \in \mathcal{N}(n)} p_{(l)}^{n,e} = 1$. Here $\mathbf{p}_{(l,k)}^n$ (row $n$ of $\mathbf{P}_{(l,k)}$) denotes node $n$'s all coefficients, which are computed by $\tag{3}$

$$\mathbf{p}_{(l,k)}^n = \phi \left( \mathbf{M} \cdot \left[ \mathbf{Z}_{(l-1)} \mathbf{W}_{(l,k)} \mathbf{N}_{(l,k)}, u_n \cdot \mathbf{1}_N \right] \mathbf{1}_2 \right) \in \mathbb{R}^N, \tag{4}$$

where $\phi$ is a softmax function, used to calculate the importance weight from other nodes. The $\mathbf{N}_{(l,k)} \in \mathbb{R}^{D_l \times 1}$ is a convolutional filter with filter size equal to $1 \times 1$ and output channel size equal to 1. The $u_n$ is the $n$'s entry of $\mathbf{Z}_{(l-1)} \mathbf{W}_{(l,k)} \mathbf{N}_{(l,k)}$. And $\mathbf{M}$ is a mask matrix with $\mathbf{M}_{n,e} = 1$ for $n \in [N]$ and $e \in \mathcal{N}(n)$, and 0 otherwise, and $\cdot$ is the element-wise product, $\mathbf{1}_N$ is the size $N$ vector with all entries equal to 1. In the paper by Veličković et al. (2017), the proposed attention mechanism $a$ consists of the following steps: firstly apply another convolutional filter $\mathbf{N}$ to the $\mathbf{x}^\top \mathbf{W}$, secondly sum up these representations to get the attention coefficients—$e_{i,j} = \mathbf{x}_i^\top \mathbf{W} \mathbf{N} + \mathbf{x}_j^\top \mathbf{W} \mathbf{N}$. We integrate the result of this whole process into an attention matrix $\mathbf{P}_{(l)}$, which functions exactly as the attention mechanism in the original architecture introduced by Veličković et al. (2017). Since the sum of row elements of the $\mathbf{P}$ equals one, we have the following property hold for $\mathbf{P}_{(l)}$: $\left\| \mathbf{P}_{(l)} \right\|_F \leq N$, i.e., the Forbunis norm of $\mathbf{P}_{(l)}$ is bounded by the total number of nodes.

---

[1] In the final layer, we have $\mathbf{P}_{(L)} \in \mathbb{R}^{M \times N}$, since we obtain output for the nodes in $\mathcal{M}$.

## 2.4 THE EMPIRICAL RISK FRAMEWORK FOR MTSF

We first introduce function spaces of GATs for MTSF. Let the $\mathcal{X}$ and $\mathcal{Y}$ be the feature and true label spaces, respectively, and $Q$ an unknown distribution over $\mathcal{X} \times \mathcal{Y}$. Let $\mathcal{F} \subset \mathcal{V}^{\mathcal{X}}$ be the hypothesis class for predictions, where $\mathcal{V}$ is another space that might be different from $\mathcal{Y}$. In our paper, we let the function space $\mathcal{F}$ be the space of our GAT classes that contains the GAT functions $f$. We defer the detailed definitions of $\mathcal{F}$ in later sections, i.e. in terms of Weights-bounded GATs for the MTSF problem, see the definition 15.

Given $\mathcal{F}$, $\mathcal{X}$, and $\mathcal{Y}$, we let $g : \mathcal{V} \times \mathcal{Y} \rightarrow [0, B]$ be the loss function defined over $\mathcal{F}$. We assume that $g$ is bounded, i.e., the range of loss is $[0, B]$. Additionally, we require $B = 1$ (if not, we can scale the loss function) without loss of generality.

We also introduce the function class $g_{\mathcal{F}} \subset [0, B]^{\mathcal{X} \times \mathcal{Y}}$ by composing the functions in $\mathcal{F}$ with $g(\cdot, \cdot)$, i.e., $g_{\mathcal{F}} = \{(\mathbf{x}, \mathbf{y}) \mapsto g(f(\mathbf{x}), \mathbf{y}) : f \in \mathcal{F}\}$.

For any risk function $g$ defined over $\mathcal{F}$, given the training set $S = \{(\mathbf{x}_1, \mathbf{y}_1), \ldots, (\mathbf{x}_M, \mathbf{y}_M)\}$ which includes $M$ i.i.d samples from $\mathcal{X} \times \mathcal{Y}$ according to distribution $Q$, the expected/population risk $\mathbf{E}(f)$ and the empirical risk function $\hat{\mathbf{E}}(f)$ are defined as:

$$\mathbf{E}(f) = \mathop{\mathbb{E}}_{(\mathbf{x}, \mathbf{y}) \sim Q} [g(f(\mathbf{x}), \mathbf{y})], f \in \mathcal{F}. \quad (5) \qquad \hat{\mathbf{E}}(f) = \frac{1}{M} \sum_{j=1}^{M} g(f(\mathbf{x}_j), \mathbf{y}_j). \quad (6)$$

A predictor $f \in \mathcal{F}$ can be generalized if for any $\delta > 0$, $\limsup_{|S|=M \rightarrow \infty} \hat{\mathbf{E}}(f) \rightarrow \mathbf{E}(f)$ a.s.

A predictor with a generalization guarantee is closely related to the complexity of its hypothesis space. In that sense, the generalization error bound for $\mathcal{F}$ is characterized by the condition where $\mathbf{E}(f)$ is bounded by the summation of $\hat{\mathbf{E}}(f)$, the ERC that is generally the dominating term, and an error function associated with the confidence of the bound and the sample size $M$.

## 2.5 THE RADEMACHER COMPLEXITY

Suppose $\mathcal{F} = \{f : \mathbf{x} \mapsto f(\mathbf{x})\}$ is a model space. We define the ERC $\mathcal{R}(\mathcal{F})$ and Rademacher complexity $\mathcal{R}_S(\mathcal{F})$ as

$$\mathcal{R}(\mathcal{F}) = \mathop{\mathbb{E}}_{\epsilon} \left[ \frac{1}{M} \sup_{f \in \mathcal{F}} \left| \sum_{j=1}^{M} \epsilon_j f(\mathbf{x}_j) \mid \mathbf{x}_1, \ldots, \mathbf{x}_N \right| \right], \quad (7) \qquad \mathcal{R}_S(\mathcal{F}) = \mathop{\mathbb{E}}_{S \sim Q} \mathcal{R}(\mathcal{F}), \quad (8)$$

where $\{\epsilon_1, \cdots, \epsilon_M\}$ are i.i.d. Rademacher variables satisfying $\mathbb{P}(\epsilon_j = 1) = \mathbb{P}(\epsilon_j = -1) = 1/2$.

# 3 GENERALIZATION BOUND FOR THE GAT MODEL

## 3.1 NOTATION

We use bold-faced letters to denote vectors and capital letters to denote matrices or fixed parameters (which should be clear from the context). Given a vector $\mathbf{w} \in \mathbb{R}^D$, $\|\mathbf{w}\|$ refers to the Euclidean norm, and for $p \geq 1$, $\|\mathbf{w}\|_p = \left( \sum_{i=1}^{F} |w_i|^p \right)^{1/p}$ refers to the $L_p$ norm. For a matrix $\mathbf{W}$, $\|\mathbf{W}\|_F$ refers to the Frobenius norm, $\|\mathbf{W}\|_F = \sqrt{\sum_i \sum_j |w^{i,j}|^2}$. A function $f : \mathbb{R}^n \rightarrow \mathbb{R}^m$ is L-Lipschitz, $L \geq 0$, if $\|f(a) - f(b)\| \leq L \|a - b\|$ for all $a, b \in \mathbb{R}^n$. We use standard big-O notation, with $\Omega(\cdot)$, $\Theta(\cdot)$, and $\mathcal{O}(\cdot)$ hiding constants.

## 3.2 FUNCTION CLASS OF GATS

Given the inputs $\mathbf{X} = (\mathbf{x}_i, \ldots, \mathbf{x}_N)$ as multiple time series with each $\mathbf{x}_i$ as input feature for node $n_i$, the class of 2-layer GATs for MTSF $f$ maps $\mathbf{x}$ to the output $f(\mathbf{x})$ that represents a C-step-away prediction expressed in equation 2. And We consider a subset of such class requiring each $f$ with a bounded weight norm, expressed as

$$\mathcal{F} = \left\{ \mathbf{x} \mapsto f(\mathbf{x}_j) \in \mathbb{R}^C; \left\| \mathbf{W}_{(1,k)} \right\|_F \leq M_1, \left\| \mathbf{w}_{(2)}^c \right\| \leq M_2, j \in [M] \right\}, \quad (9)$$

here we use $f(\mathbf{x}_j)$ to mean the output of $f$ corresponding to node $j$, where $j \in [M]$, and we know the true label $\mathbf{y}_j$ of this node.

Furthermore, we also provide a model space $\mathcal{F}^c \subset \mathbb{R}^{\mathcal{X}}$ with a single dimensional output that corresponds to the $c$-th component of model output from $f(\mathbf{x})$ for the $c$-th time step, expressed as

$$\mathcal{F}^c = \left\{ \mathbf{x} \mapsto f(\mathbf{x}_j)^c, \left\| \mathbf{W}_{(1,k)} \right\|_F \leq M_1, \left\| \mathbf{w}^c_{(2)} \right\| \leq M_2, c \in [C], j \in [M] \right\}. \tag{10}$$

### 3.3 An Upper Bound of Rademacher Complexity of GAT Class

Here we first provide an upper bound of ERC of GAT class $\mathcal{F}^c$ for single dimensional output of MTSF.

**Theorem 1** (Upper Bound of ERC of GAT class $\mathcal{F}^c$ for MTSF). *Let the activation function $\sigma(\cdot)$ be $L_\sigma$-Lipschitz continuous, and also satisfy $\sigma(0) = 0$ and $\sigma(\alpha z) = \alpha \sigma(z)$ for all $\alpha \geq 0$. Assume the $L_2$-norm of the feature vector $\mathbf{x}$ comes from a bounded domain $\mathcal{X} = \{ \mathbf{x} : \| \mathbf{x} \| \leq B \}$. Assume the Frobenius norm of every weight matrix in the first layer of the GAT class is bounded, namely, $\left\| \mathbf{W}_{(1,k)} \right\|_F \leq M_1$ with some constant $M_1 > 0$ for every $k$. Also, the norm of the weight vector of the second layer of the GATs is also bounded, $\left\| \mathbf{w}^c_{(2)} \right\| \leq M_2$, where $c \in [C]$, with some constant $M_2 > 0$. Let $\mathcal{N}_i$ denote the neighborhood of node $i$ (including $i$), let the number of neighbors of each node is equal to each other, namely, for some common constant $N_e \in \mathbb{N}^+$, assume $N_e := |\mathcal{N}_i|$ for all node $i \in \mathcal{N}$, furthermore, we consider the most general formulation, which allows every node to attend on every other node, i.e., $N_e = N$. Then let $\mathcal{R}(\mathcal{F}^c)$ be the ERC defined in the definition 7 for GAT class $\mathcal{F}^c$ in the definition 17, given the $M$ sized input set $\{\mathbf{x}_1, \ldots, \mathbf{x}_M\}$, then we have*

$$\mathcal{R}(\mathcal{F}^c) = \mathcal{O}(L_\sigma B K (N)^{3/2} M^{-1/2} M_1 M_2).$$

We see that this bound has a polynomial dependence on the $N$, the total number of nodes. The $N$ appears here due to the fact that we consider all the nodes as neighbors, which can be replaced by the $N_e$, the number of neighbors. A small $N_e < N$ can result in a potentially smaller bound. The proof details can be found in §A.

### 3.4 Generalization Error Bound of the GAT Class for MTSF

In this section, we will give the final generalization error bound of the GAT class for MTSF. The formal result is in the following theorem and the proof is in §B.

**Theorem 2.** *Define the hypothesis class $\mathcal{F}$ as the definition 15. We suppose $g$ is Lipschitz with constant $L_g$. Then for any $\delta \in (0, 1)$, with probability at least $1 - \delta$, for all $f \in \mathcal{F}$, we have*

$$\mathbf{E}(f) \leq \hat{\mathbf{E}}(f) + 2\sqrt{2} C L_g \mathcal{R}(\mathcal{F}^c) + 3\sqrt{\frac{\ln(2/\delta)}{2M}},$$

*where we have $\mathcal{R}(\mathcal{F}^c) = \mathcal{O}(L_\sigma B K (N)^{3/2} M^{-1/2} M_1 M_2)$ from Theorem 1.*

## 4 Extension to GAT Class with Layers $L > 2$

Now we extend the analysis to GATs with more than two layers for MTSF and provide corresponding generalization error bounds. Here, the proof is done by a simple induction argument using the "peeling-off" technique employed for Rademacher complexity bounds for neural networks. The output of a $L$-layer GATs represents a multi-step-away prediction shown in expression 1.

We define the function class over $\mathcal{M}$, according to the definition of the GATs network, with

$$\mathcal{F} = \left\{ \mathbf{x} \mapsto f(\mathbf{x}_j) \in \mathbb{R}^C : \left\| \mathbf{W}_{(1,k)} \right\|_F \leq M_1, \ldots, \left\| \mathbf{w}^c_{(L)} \right\| \leq M_L, c \in [C], j \in [M] \right\}. \tag{11}$$

Furthermore, we require that the Frobenius norm of every weight matrix in every layer of the GAT class is bounded, namely, for any $l \in [L]$, $\left\| \mathbf{W}_{(l,k)} \right\|_F \leq M_l$ with some constant $M_l > 0$ for every $k$.

Also, we have single dimensional GAT function space defined as

$$\mathcal{F}^c = \{\mathbf{x} \mapsto f(\mathbf{x}_j)^c : f \in \mathcal{F}, c \in [C], j \in [M]\}. \tag{12}$$

Also, the output up to layer $l$, $l \in [L-1]$, has the format as,

$$\left[f_l(\mathbf{x}_1)^\top, \ldots, f_l(\mathbf{x}_M)^\top\right]^\top = \oplus_{k_l=1}^{K_l} \sigma\left(\mathbf{P}_{(l)} \cdots \oplus_{k_2=1}^{K_2} \sigma\left(\mathbf{P}_{(2)} \oplus_{k_1=1}^{K_1} \sigma\left(\mathbf{P}_{(1,k)} \mathbf{X} \mathbf{W}_{(1,k)}\right) \mathbf{W}_{(2)}\right) \cdots \mathbf{W}_{(l)}\right) \in \mathbb{R}^{D_l}. \tag{13}$$

Thus, we define a layer-wised class of functions as

$$\mathcal{F}_l = \left\{f_l : \mathbf{x} \mapsto f_l(\mathbf{x}_j) \in \mathbb{R}^{D_l}, \left\|\mathbf{W}_{(1,k)}\right\|_F \leq M_1, \ldots, \left\|\mathbf{W}_{(l,k)}\right\|_F \leq M_l, j \in [M]\right\}. \tag{14}$$

We provide an upper bound of ERC of GAT class $\mathcal{F}^c$ with $L$ layers with proof details in §C

**Theorem 3** (Upper Bound of ERC of GAT class $\mathcal{F}^c$ with $L$ layers). *Let all the assumptions from Theorem 1 be fulfilled. Furthermore, let the Frobenius norm of every weight matrix in the first $L-1$ layers of the GATs be bounded, namely, $\left\|\mathbf{W}_{(l,k)}\right\|_F \leq M_l$ with some constant $M_l > 0$ for every $k$. Also, the norm of the weight vector of the last layer is also bounded, $\left\|\mathbf{w}_{(L)}^c\right\| \leq M_L$, where $c \in [C]$, with some constant $M_L > 0$. Let $\mathcal{R}(\mathcal{F}^c)$ be the ERC defined in Equation 7 for GAT class $\mathcal{F}^c$ in the definition 12, given the $M$ sized input set $\{\mathbf{x}_1, \ldots, \mathbf{x}_M\}$, then we have*

$$\mathcal{R}(\mathcal{F}^c) = \mathcal{O}\left((4L_\sigma K)^{L-1} \prod_{l=1}^{L} M_l (B N^{L-1/2} M^{-1/2})\right).$$

Based on the theorem 3, we provide the generalization error bounds for the GAT class with more than two layers in the following theorem.

**Theorem 4** (Generalization Error Bounds for the GAT Class with More than Two Layers). *Given any real $\delta \in (0,1)$, with probability at least $1 - \delta$, for any $f \in \mathcal{F}$ defined in the definition 11, which contains $L$-layer GATs, and given the $M$ sized training data set $S = \{(\mathbf{x}_1, \mathbf{y}_1), \ldots, (\mathbf{x}_M, \mathbf{y}_M)\}$, we have the generalization error for MTSF is upper bounded as*

$$\mathbf{E}(f) \leq \hat{\mathbf{E}}(f) + 2\sqrt{2} C L_g \mathcal{R}(\mathcal{F}^c) + 3\sqrt{\frac{\ln(2/\delta)}{2M}},$$

*where $\mathcal{R}(\mathcal{F}^c)$ is calculated in theorem 3.*

To give the generalization error bound of a deep GAT, we firstly derive its ERC. To bound the ERC, we apply the layer-peeling strategy that the ERC of $L$-layer networks is expressed by a factor multiplied by the ERC of $L-1$-layer networks. Specifically, we consider this factor as the matrix Frobenius norm, and our current bounds scale with the product of these norms as the layer size increases. However, the number of attention heads and the number of attention neighbors appear as the polynomial terms with an order roughly equal to the number of layers $L$. The bound has an exponential dependence on the network depth.

The generalization error bound in Theorem 4 implies that the following attempts can be taken to reduce the generalization error: i) increase the training samples, ii) minimize the empirical loss, and iii) design the neural network carefully to achieve a proper hypothesis class. Increasing the complexity of the hypothesis class can decrease the approximation error but also increase the estimation error due to a large ERC, which leads to undesired test performance in a practical task. In the next section, we will empirically show how structural components of GATs related to complexity could affect the test performance for a MTSF task, providing empirical support for our theoretical findings.

## 5 EXPERIMENT DETAILS

Our goal is to show the relationship between the upper bound of the generalization error of the GAT model and variables in the ERC, including the number of attention heads, the number of neighbors, the Frobenius norm of model weights, the number of model layers, the norm of inputs, and the number of labeled nodes. We use the daily stock price data from Nasdaq and NYSE. The multivariate time series have about 1500 stocks. For each stock, the one-week historical data is used to predict future returns. For every stock, we assume that every feature is i.i.d. per day. By default, we use a three-layer (input layer, hidden layer, and output layer) GAT to do single-step forecasting on the

returns of each stock. For each variable, we try its various values with all other variables's values fixed. We repeat the experiment **20** times and report the loss on the out-of-sample test dataset to represent its generalization error. We use minimum square error (MSE) as evaluation metrics [2].

# 6 EXPERIMENT RESULTS

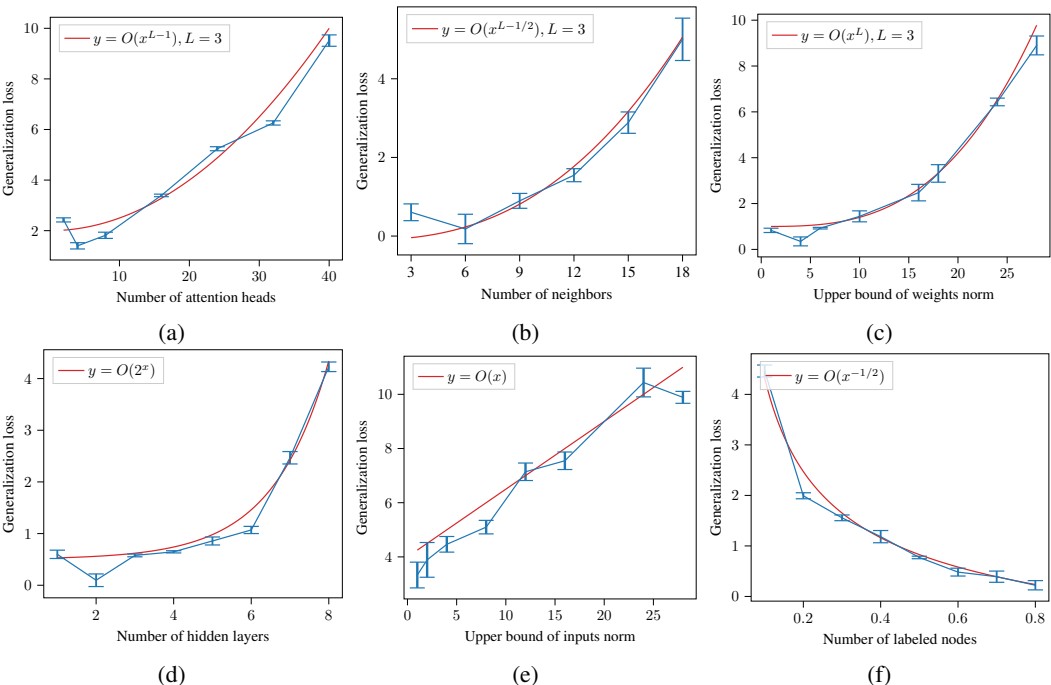

(a)  (b)  (c)

(d)  (e)  (f)

Figure 1: Experiment results on six variables in the ERC. We run the experiment 20 times and obtain a standard deviation of the generalization error. (a) relationship between test loss and the number of attention heads. (b) relationship between test loss and the number of neighbors. (c) relationship between test loss and the upper bound of weight norm. (d) relationship between test loss and the number of (hidden) layers. (e) relationship between test loss and the upper bound of input norm. (f) relationship between test loss and the number of labeled nodes. The red line is a possible theoretical upper bound. The plots show that all test losses generally conform to the big O of the theoretical upper bound.

**Number of Attention Heads - $K$.** For three-layer GATs, Theorem 3 indicates that with the increasing number of attention heads in the attention layer, the upper bound of ERC is $y = O(K^2)$. The experiment results in Figure 1 shows test error beginning to increase quadratically after some values of $K$. This is consistent with our theoretical results on the generalization error bound.

**Number of Neighbors - $N_e$.** For three-layer GATs, Theorem 3 indicates that with the increasing number of neighbors considered for the attention operation, the upper bound of ERC is $y = O(N_e^{L-1/2})$. Figure 1 demonstrates that as the $N_e$ increases, the test error conforms to this theoretical error bound. But it is noteworthy that when the number is too small, the loss is also high. It is possible that at certain range, the influence of the information loss due to the limited number of neighbors is dominant.

**Norm of Weight Matrix - $M_l$.** We also have an empirical evaluation on the relationship between the generalization error bound and the Frobenius norm of GAT models' weights. Theorem 3 indicates that with the increasing bound of the Frobenius norm, the upper bound of ERC increases polynomially. The experiment results in Figure 1 corroborate the conclusion. When the weight norm increases, the

---

[2]It is known that MSE loss is not Lipschitz continuous over all $\mathcal{Y}$, however, since we consider a finite hypothesis class satisfying bounded input and weights conditions, the output of GATs is bounded, thus, the MSE is a locally Lipschitz continuous function.

generalization error first decreases, then increases. The reason for the initial decrease is because the bound on the norms is so small that it severely prevents the weights from having enough amount of updates. Thus, the scales (norms) of the weight matrices should be neither too large (induces large generalization error) nor too small (harms weights' updates) and choosing proper scales is important in practice as the current work has shown Li et al. (2018).

**Number of Layers (Model Depth) - $L$.**   Theorem 3 and Theorem 1 indicate that the upper bound of ERC increases exponentially with an increasing number of layers, suggesting that the number of layers has negative impact on the test performance of GATs for MTSF. However, Sun et al. (2016) reported that deeper nets which have larger representation power are able to fit training data better and achieve smaller empirical error. This observation indicates a positive impact of the number of layers' positive on the test performance for MTSF. Our empirical results are consistent with the above discussions about the double-edged impacts. The experiment results are shown in Figure 1 and indicate that as the number of layers increases, the test error first decreases, and then increases. This observation indicates that if the ERC increases quickly, the representation power cannot compensate for the negative impact of the increased number of layers.

**Norm of Inputs - $B$.**   The experiment results in Figure 1 show that when the input norm is greater than certain values, the generalization loss of the GAT has a linear relationship with the upper bound of the input norm. This empirical observation aligns with Theorem 3.

**Number of Labeled Nodes - $M$.**   Theorem 3 also indicates that when the number of labeled nodes increases, the generalization loss of the GAT decreases and the upper bound of the generalization loss is $y = O(M^{-1/2})$. The empirical results in Figure 1 confirm this relationship.

We include the details about how we control the above six variables in D.2. In addition, we conduct experiments on two-layer GATs to show that empirical results are also consistent with error bounds in Theorem 1. Results and discussions are in §E.

It is noteworthy that Figure 1 also shows that the test error decreases initially, then starts to increase as Theorem 3 suggests. The inconsistency at the beginning could be brought about by other factors that affect the test error. As seen in the Theorem 4, in addition to the ERC, the training error also contributes to the upper bound. Nevertheless, the ERC becomes dominant by increasing the number of attention heads, the number of neighbors, the upper bound of weights norm, and the number of hidden layers, indicating the importance of a proper design of neural networks for MTSF to guarantee a smaller test error. Among all the above factors that affect the generalization error, since the order of the upper bound of the weights norm is the highest, properly controlling the upper bound of the weights norm is especially significant.

As reported in the previous literature (Wu et al., 2021), increasing the complexity hypothesis class in terms of larger weight matrices bounds could decrease the approximation error, but may also increase the estimation error, which corresponds to the second term of RHS in Theorem 4.

Therefore, in practical training process, we generally start with a simple neural network and gradually increase its complexity in terms of larger weight matrix bounds to improve the test performance, and the bound can be a tuning parameter in our model. We call it weight control.

To show the meanings of weight control, we further developed an improved version of the GAT called Weights-bounded GAT. And we conduct an additional experiment to demonstrate its improvement over the vanilla GAT. As the name suggests, the weight Frobenius norm of each layer is bounded by a hyperparameter. We compare the Weights-bounded GAT with the vanilla GAT using test loss on the above stock return forecasting task. We repeat the same train-test pipeline 20 times and collect 20 test losses for each model. As Figure 2 shows, the Weights-bounded GAT performs much better than the vanilla GAT.

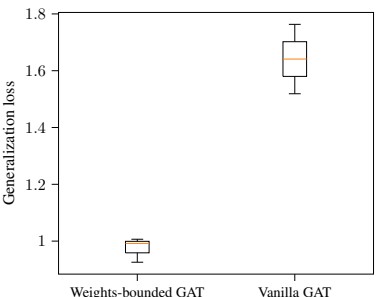

Figure 2: Test loss of two types of GATs. The train-test pipeline runs 20 times over 20 random seeds. The Weights-bounded GAT has a better test loss than the vanilla GAT regarding the minimum, maximum, first quantile, third quantile and the median of the test loss.

While the neural network training process can only minimize the training error and has no guarantee on generalization error, this generalization error bound analyzed in this study is more useful in the sense that it provides a solid guideline on GAT structure design to improve the models generalization performance for MTSF.

## 7 ETHICS STATEMENT

We have read the code of ethics carefully and ensured that our paper conforms to them.

## 8 REPRODUCIBILITY STATEMENT

We make our best effort to ensure the reproducibility of the paper's experiments and provide clear guidelines. More specifically, we include detailed experiment setup information in D.1. Besides, we give detailed description on the architecture of the neural network we use and we explain how we control and adjust different variables in D.2. We list all the important hyperparameters and their default values in D.3. We give the website link for the dataset we use in D.4. There is no extra data processing steps to be conducted. All the work is done in the code in an end-to-end manner. We also give the link to the source code in D.5.

We explain all assumptions of our theorems (Theorem 1, 2, 3, 4) clearly, and provide the complete proof of the theorems in the appendix §A, §B, and §C.

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

**APPENDICES TO MULTIVARIATE TIME SERIES FORECASTING BY GRAPH ATTENTION NETWORKS WITH THEORETICAL GUARANTEES**

## A    PROOF OF THEOREM 1

Here we will derive the upper bound of ERC of two-layer GATs with on-dimensional output for the single-time-step prediction.

As mentioned in section 2.3, our GATs model's second layer $l_{(2)}$ uses $\mathbf{Z}_{(1)} \in \mathbb{R}^{N \times KD_1}$ as input, and outputs $\mathbf{Z}_{(2)} \in \mathbb{R}^{M \times C}$. The input to the first layer $l_{(1)}$ is a set of node features

$$\mathbf{X} = (\mathbf{x}_1, \ldots, \mathbf{x}_N),$$

, where each $\mathbf{x}_i \in \mathbb{R}^D$, $D$ is the number of features in each node. The first layer produces $\mathbf{Z}_{(1)}$ with each $\mathbf{z}_{(1)} \in \mathbb{R}^{KD_1}$. The $\mathbf{Z}_{(1)}$ is composed by concatenation of $K$ outputs from the $K$ identically structured attention layers $l_{(1,k)}$, with each output denoted as

$$\mathbf{H}_{(1,k)} = \sigma(\mathbf{P}_{(1,k)} \mathbf{X} \mathbf{W}_{(1,k)})$$

Now we write the output vector of GATs for all classes of node $j$, $j \in [M]$ as,

$$z_{(2)}^j = \sum_{kr=1}^{KD_1} \mathbf{w}_{(2)}^{kr} \sum_{t=1}^{N} p_{(2)}^{j,t} \cdot \sum_{k=1}^{K} \sigma \left( \sum_{d=1}^{D} w_{(1,k)}^{d,r} \sum_{i=1}^{N} p_{(1,k)}^{t,i} x^{i,d} \right).$$

Here we use index notation $kr \in [KD_1]$ because we already have indexes $k \in [K]$ and $r \in [D_1]$. And it is easy to show that above output can be easily written as in vector format:

$$z_{(2)}^j = \sum_{kr=1}^{KD_1} \mathbf{w}_{(2)}^{kr} \sum_{t=1}^{N} p_{(2)}^{j,t} \cdot \sum_{k=1}^{K} \sigma \left( \sum_{i=1}^{N} p_{(1,k)}^{t,i} \left\langle \mathbf{w}_{(1,k)}^r, \mathbf{x}_i \right\rangle \right),$$

where $\mathbf{w}_{(1,k)}^r$ represents the column $r$ of $\mathbf{W}_{(1,k)}$, $\mathbf{w}_{(2)}^{kr}$ represents the row $kr$ of $\mathbf{W}_{(2)}$, and $\mathbf{w}_{(2)}^c$ represents the column $c$ of $\mathbf{W}_{(2)}$. Then the class of functions defined over the subset of the node set $\mathcal{M}$, $|\mathcal{M}| = M$, will be

$$\mathcal{F} = \left\{ \mathbf{x} \mapsto f(\mathbf{x}_j) = \sum_{kr=1}^{KD_1} \mathbf{w}_{(2)}^{kr} \sum_{t=1}^{N} p_{(2)}^{n,t} \cdot \sum_{k=1}^{K} \sigma \left( \sum_{i=1}^{N} p_{(1,k)}^{t,i} \left\langle \mathbf{w}_{(1,k)}^r, \mathbf{x}_i \right\rangle \right) \in \mathbb{R}^C; \quad (15) \right.$$

$$\left. \left\| \mathbf{W}_{(1,k)} \right\|_F \leq M_1, \left\| \mathbf{w}_{(2)}^c \right\| \leq M_2, j \in [M] \right\}, \quad (16)$$

Furthermore, we also provide a model space with a single dimensional output that corresponds to the $c$-th component of model output from $f(\mathbf{x})$ for the $c$-th time step. Now we write the output of node $n$ of time-step $c$ of $l_{(2)}$ as

$$z_{(2)}^{j,c} = \sum_{kr=1}^{KD_1} w_{(2)}^{kr,c} \sum_{t=1}^{N} p_{(2)}^{n,t} \cdot \sum_{k=1}^{K} \sigma \left( \sum_{d=1}^{F} w_{(1,k)}^{d,r} \sum_{i=1}^{N} p_{(1,k)}^{t,i} x_i^d \right),$$

and its vector format is

$$z_{(2)}^{j,c} = \sum_{kr=1}^{KD_1} w_{(2)}^{kr,c} \sum_{t=1}^{N} p_{(2)}^{n,t} \cdot \sum_{k=1}^{K} \sigma \left( \sum_{i=1}^{N} p_{(1,k)}^{t,i} \left\langle \mathbf{w}_{(1,k)}^r, \mathbf{x}_i \right\rangle \right).$$

Now let hypothesis class $\mathcal{F}^c \subset \mathbb{R}^{\mathcal{X}}$ be a set of functions on $\mathbf{x} \in \mathcal{X}$. Specifically, we have such single dimensional GAT function space defined as

$$\mathcal{F}^c = \left\{ \mathbf{x} \mapsto f(\mathbf{x}_j)^c = \sum_{kr=1}^{KD_1} w_{(2)}^{kr,c} \sum_{t=1}^{N} p_{(2)}^{j,t} \cdot \sum_{k=1}^{K} \sigma \left( \sum_{i=1}^{N} p_{(1,k)}^{t,i} \left\langle \mathbf{w}_{(1,k)}^r, \mathbf{x}_i \right\rangle \right); \right.$$

$$\left. \left\| \mathbf{W}_{(1,k)} \right\|_F \leq M_1, \left\| \mathbf{w}_{(2)}^c \right\| \leq M_2, c \in [C], j \in [M] \right\}. \quad (17)$$

We have the first layer's matrix to be

$$\mathbf{W}_{(1,k)} = \left[\mathbf{w}_{(1,k)}^1, \dots, \mathbf{w}_{(1,k)}^{D_1}\right].$$

The output of first layer $l_1$ is

$$\mathbf{Z}_{(1)} = \left[\mathbf{h}_{(1,k)}^{1\top}, \mathbf{h}_{(1,k)}^{2\top}, \dots, \mathbf{h}_{(1,k)}^{N\top}\right]^\top \in \mathbb{R}^{N \times KD_1}.$$

We then further write the output of attention head $k$ of first layer $l_{(1)}$ as

$$\mathbf{H}_{(1,k)} = \left[\mathbf{h}_{(1,k,1)}^\top, \mathbf{h}_{(1,k,2)}^\top, \dots, \mathbf{h}_{(1,k,N)}^\top\right]^\top \in \mathbb{R}^{N \times D_1}.$$

By the concatenation relationship, we have the row $t$ of $\mathbf{Z}_{(1)}$ to be

$$\mathbf{z}_{(1)}^t = \left[\mathbf{h}_{(1,1,t)}^\top, \mathbf{h}_{(1,2,t)}^\top, \dots, \mathbf{h}_{(1,k,t)}^\top\right] \in \mathbb{R}^{KD_1}. \tag{18}$$

And we define each $\mathbf{h}_{(1,k,t)}$ as

$$\mathbf{h}_{(1,k,t)} = \left[h_{(1,k,t)}^1, h_{(1,k,t)}^2, \dots, h_{(1,k,t)}^{D_1}\right]^\top \in \mathbb{R}^{D_1},$$

with each $h_{(1,k,t)}^r \in \mathbb{R}$ defined as

$$h_{(1,k,t)}^r = \sigma\left(\sum_{i=1}^N p_{(1,k)}^{t,i}\left\langle \mathbf{w}_{(1,k)}^r, \mathbf{x}_i\right\rangle\right).$$

Then the output of node $j$ of class $c$ can be re-written as

$$z_{(2)}^{j,c} = \phi\left(\sum_{t=1}^N p_{(2)}^{j,t} \cdot \left\langle \mathbf{z}_{(1)}^t, \mathbf{w}_{(2)}^c\right\rangle\right),$$

where $\mathbf{w}_{(2)}^c$ represents column $c$ of $\mathbf{W}_{(2)}$.

According to the definition of $\mathcal{F}^c$ in definition 17, and ERC, we have

$$
\begin{aligned}
\mathcal{R}(\mathcal{F}^c) &= \mathbb{E}_\epsilon\left[\frac{1}{M}\sup_{f\in\mathcal{F}}\left|\sum_{j=1}^M \epsilon_j f(\mathbf{x}_j)^c\right|\right] = \mathbb{E}_\epsilon\left[\frac{1}{M}\sup_{\substack{\|\mathbf{W}_{(1,k)}\|_F\leq M_1 \\ \|\mathbf{w}_{(2)}^c\|\leq M_2}}\left|\sum_{j=1}^M \epsilon_j \sum_{t=1}^N p_{(2)}^{j,t}\cdot\left\langle \mathbf{z}_{(1)}^t, \mathbf{w}_{(2)}^c\right\rangle\right|\right] \\
&= \mathbb{E}_\epsilon\left[\frac{1}{M}\sup_{\substack{\|\mathbf{W}_{(1,k)}\|_F\leq M_1 \\ \|\mathbf{w}_{(2)}^c\|\leq M_2}}\left|\left\langle \sum_{j=1}^M \epsilon_j \sum_{t=1}^N p_{(2)}^{j,t}\cdot\mathbf{z}_{(1)}^t, \mathbf{w}_{(2)}^c\right\rangle\right|\right] \\
&\leq \frac{M_2}{M}\mathbb{E}_\epsilon\left[\sup_{\|\mathbf{W}_{(1,k)}\|_F\leq M_1}\left\|\sum_{j=1}^M \epsilon_j \sum_{t=1}^N p_{(2)}^{j,t}\cdot\mathbf{z}_{(1)}^t\right\|\right]
\end{aligned}
$$

The inequality comes from the Cauchy-Schwartz inequality.

We further unfold the expectation, we have

$$
\begin{aligned}
&\mathbb{E}_\epsilon\left[\sup_{\|\mathbf{W}_{(1,k)}\|_F\leq M_1}\left\|\sum_{j=1}^M \epsilon_j \sum_{t=1}^N p_{(2)}^{j,t}\cdot\mathbf{z}_{(1)}^t\right\|\right] \\
&= \mathbb{E}_\epsilon\left[\sup_{\|\mathbf{w}_{(1,k)}\|=M_1}\sum_{k=1}^K\left|\sum_{j=1}^M \epsilon_j \sum_{t=1}^N p_{(2)}^{j,t}\sigma\left(\sum_{i=1}^N p_{(1,k)}^{t,i}\left\langle \mathbf{w}_{(1,k)}, \mathbf{x}_i\right\rangle\right)\right|\right],
\end{aligned}
\tag{19}
$$

The equality is due to the following derivation:

$$
\left\| \sum_{j=1}^{M} \epsilon_j \sum_{t=1}^{N} p_{(2)}^{i,t} \cdot \mathbf{z}_{(1)}^t \right\|^2
$$

$$
= \sum_{kr=1}^{KD_1} \left( \sum_{j=1}^{M} \epsilon_j \sum_{t=1}^{N} p_{(2)}^{j,t} h_{(1,k,t)}^r \right)^2 = \sum_{k=1}^{K} \sum_{r=1}^{D_1} \left( \sum_{j=1}^{M} \epsilon_j \sum_{t=1}^{N} p_{(2)}^{j,t} h_{(1,k,t)}^r \right)^2
$$

$$
= \sum_{k=1}^{K} \sum_{r=1}^{D_1} \left( \sum_{j=1}^{M} \epsilon_j \sum_{t=1}^{N} p_{(2)}^{j,t} \sigma \left( \sum_{i=1}^{N} p_{(1,k)}^{t,i} \left\langle \mathbf{w}_{(1,k)}^r, \mathbf{x}_i \right\rangle \right) \right)^2
$$

For a fixed $k$-th attention head, we let the $\mathbf{w}_{(1,k)}^1, \mathbf{w}_{(1,k)}^2, \ldots, \mathbf{w}_{(1,k)}^{D_1}$ be the the columns of $\mathbf{W}_{(1,k)}$, then, by positive homogeneity of $\sigma$, we have

$$
\sum_{k=1}^{K} \sum_{r=1}^{D_1} \left( \sum_{j=1}^{M} \epsilon_j \sum_{t=1}^{N} p_{(2)}^{j,t} \sigma \left( \sum_{i=1}^{N} p_{(1,k)}^{t,i} \left\langle \mathbf{w}_{(1,k)}^r, \mathbf{x}_i \right\rangle \right) \right)^2
$$

$$
= \sum_{k=1}^{K} \sum_{r=1}^{D_1} \left\| \mathbf{w}_{(1,k)}^r \right\|^2 \left( \sum_{j=1}^{M} \epsilon_j \sum_{t=1}^{N} p_{(2)}^{j,t} \sigma \left( \sum_{i=1}^{N} p_{(1,k)}^{t,i} \left\langle \frac{\mathbf{w}_{(1,k)}^r}{\left\| \mathbf{w}_{(1,k)}^r \right\|}, \mathbf{x}_i \right\rangle \right) \right)^2
$$

The supremum of this quantity over $\mathbf{w}_{(1,k)}^1, \mathbf{w}_{(1,k)}^2, \ldots, \mathbf{w}_{(1,k)}^{D_1}$ under the constraint that $\left\| \mathbf{W}_{(1,k)} \right\|_F^2 \leq M_1^2 = \sum_{r=1}^{D_1} \left\| \mathbf{w}_{(1,k)}^r \right\|^2 \leq M_1^2$ is attained when $\left\| \mathbf{w}_{(1,k)}^r \right\| = M_1$ for some $r$ and $\left\| \mathbf{w}_{(1,k)}^{r'} \right\| = 0$ for all other $r' \neq r$. In the end, only the $r$ terms remain. For simplicity of notation, we use $\mathbf{w}_{(1,k)}$ to mean that $r$'s column $\mathbf{w}_{(1,k)}^r$.

Therefore, we have

$$
\mathbb{E}_{\epsilon} \left[ \sup_{\left\| \mathbf{W}_{(1,k)} \right\|_F \leq M_1} \left\| \sum_{j=1}^{M} \epsilon_j \sum_{t=1}^{N} p_{(2)}^{j,t} \cdot \mathbf{z}_{(1)}^t \right\| \right]
$$

$$
= \mathbb{E}_{\epsilon} \left[ \sup_{\left\| \mathbf{w}_{(1,k)} \right\| = M_1} \sum_{k=1}^{K} \left| \sum_{j=1}^{M} \epsilon_j \sum_{t=1}^{N} p_{(2)}^{j,t} \sigma \left( \sum_{i=1}^{N} p_{(1,k)}^{t,i} \left\langle \mathbf{w}_{(1,k)}, \mathbf{x}_i \right\rangle \right) \right| \right]
$$

$$
\overset{(a)}{\leq} \sum_{k=1}^{K} \mathbb{E}_{\epsilon} \left[ \sup_{\left\| \mathbf{w}_{(1,k)} \right\| = M_1} \left| \sum_{j=1}^{M} \epsilon_j \sum_{t=1}^{N} p_{(2)}^{j,t} \sigma \left( \sum_{i=1}^{N} p_{(1,k)}^{t,i} \left\langle \mathbf{w}_{(1,k)}, \mathbf{x}_i \right\rangle \right) \right| \right]
$$

$$
\overset{(b)}{\leq} 2 \sum_{k=1}^{K} \mathbb{E}_{\epsilon} \left[ \sup_{\left\| \mathbf{w}_{(1,k)} \right\| = M_1} \sum_{j=1}^{M} \epsilon_j \sum_{t=1}^{N} p_{(2)}^{j,t} \sigma \left( \sum_{i=1}^{N} p_{(1,k)}^{t,i} \left\langle \mathbf{w}_{(1,k)}, \mathbf{x}_i \right\rangle \right) \right]
$$

(20)

The inequality (a) is because $\sup(\sum_i x_i) \leq \sum_i \sup(x_i)$.

For the inequality (b) above, we have

$$
\begin{aligned}
&\mathbb{E}_{\epsilon}\left[\sup_{\|\mathbf{w}_{(1,k)}\|=M_1}\left|\sum_{j=1}^{M}\epsilon_j\sum_{t=1}^{N}p_{(2)}^{j,t}\sigma\left(\sum_{i=1}^{N}p_{(1,k)}^{t,i}\left\langle\mathbf{w}_{(1,k)},\mathbf{x}_i\right\rangle\right)\right|\right]\\
&\stackrel{(a)}{=}\mathbb{E}_{\epsilon}\left[\sup_{\|\mathbf{w}_{(1,k)}\|=M_1}\left(\sum_{j=1}^{M}\epsilon_j\sum_{t=1}^{N}p_{(2)}^{j,t}\sigma\left(\sum_{i=1}^{N}p_{(1,k)}^{t,i}\left\langle\mathbf{w}_{(1,k)},\mathbf{x}_i\right\rangle\right)\right)_+ + \left(\sum_{j=1}^{M}\epsilon_j\sum_{t=1}^{N}p_{(2)}^{j,t}\sigma\left(\sum_{i=1}^{N}p_{(1,k)}^{t,i}\left\langle\mathbf{w}_{(1,k)},\mathbf{x}_i\right\rangle\right)\right)_-\right]\\
&\stackrel{(b)}{\leq}\mathbb{E}_{\epsilon}\left[\sup_{\|\mathbf{w}_{(1,k)}\|=M_1}\left(\sum_{j=1}^{M}\epsilon_j\sum_{t=1}^{N}p_{(2)}^{j,t}\sigma\left(\sum_{i=1}^{N}p_{(1,k)}^{t,i}\left\langle\mathbf{w}_{(1,k)},\mathbf{x}_i\right\rangle\right)\right)_+\right]\\
&\quad + \mathbb{E}_{\epsilon}\left[\sup_{\|\mathbf{w}_{(1,k)}\|=M_1}\left(\sum_{j=1}^{M}\epsilon_j\sum_{t=1}^{N}p_{(2)}^{j,t}\sigma\left(\sum_{i=1}^{N}p_{(1,k)}^{t,i}\left\langle\mathbf{w}_{(1,k)},\mathbf{x}_i\right\rangle\right)\right)_-\right]\\
&\stackrel{(c)}{=}2\mathbb{E}_{\epsilon}\left[\sup_{\|\mathbf{w}_{(1,k)}\|=M_1}\left(\sum_{j=1}^{M}\epsilon_j\sum_{t=1}^{N}p_{(2)}^{j,t}\sigma\left(\sum_{i=1}^{N}p_{(1,k)}^{t,i}\left\langle\mathbf{w}_{(1,k)},\mathbf{x}_i\right\rangle\right)\right)_+\right]\\
&\stackrel{(d)}{=}2\mathbb{E}_{\epsilon}\left[\left(\sup_{\|\mathbf{w}_{(1,k)}\|=M_1}\sum_{j=1}^{M}\epsilon_j\sum_{t=1}^{N}p_{(2)}^{j,t}\sigma\left(\sum_{i=1}^{N}p_{(1,k)}^{t,i}\left\langle\mathbf{w}_{(1,k)},\mathbf{x}_i\right\rangle\right)\right)_+\right]\\
&\stackrel{(e)}{=}2\mathbb{E}_{\epsilon}\left[\left(\sup_{\|\mathbf{w}_{(1,k)}\|=M_1}\sum_{j=1}^{M}\epsilon_j\sum_{t=1}^{N}p_{(2)}^{j,t}\sigma\left(\sum_{i=1}^{N}p_{(1,k)}^{t,i}\left\langle\mathbf{w}_{(1,k)},\mathbf{x}_i\right\rangle\right)\right)\right]
\end{aligned}
\tag{21}
$$

where the equality (a) above is due to $|x| = (x)_+ + (x)_-$, and the inequality (b) is due to $\sup_{A+B} \leq \sup_A + \sup_B$, and the equality (c) comes from the symmetry in the distribution of the $\epsilon_i$ random variables. The equality (d) is due to $\sup_{A_+} = (\sup_A)_+$. The equality (e) is because the supremum is non-negative, as when $\mathbf{w}_{(1,k)} = \mathbf{0}$, we can get the $\sum_{j=1}^{M}\epsilon_j\sum_{t=1}^{N}p_{(2)}^{j,t}\sigma\left(\sum_{i=1}^{N}p_{(1,k)}^{t,i}\left\langle\mathbf{w}_{(1,k)},\mathbf{x}_i\right\rangle\right) = 0$.

We then rewrite

$$
\sum_{j=1}^{M}\epsilon_j\sum_{t=1}^{N}p_{(2)}^{j,t}\sigma\left(\sum_{i=1}^{N}p_{(1,k)}^{t,i}\left\langle\mathbf{w}_{(1,k)},\mathbf{x}_i\right\rangle\right) = \sum_{t=1}^{N}\sum_{j=1}^{M}\epsilon_j p_{(2)}^{j,t}\sigma\left(\sum_{i=1}^{N}p_{(1,k)}^{t,i}\left\langle\mathbf{w}_{(1,k)},\mathbf{x}_i\right\rangle\right)
$$

By the same argument in $\sup(\sum_i x_i) \leq \sum_i \sup(x_i)$, we have

$$
\begin{aligned}
&\mathbb{E}_{\epsilon}\left[\sup_{\|\mathbf{w}_{(1,k)}\|=M_1}\sum_{j=1}^{M}\epsilon_j\sum_{t=1}^{N}p_{(2)}^{j,t}\sigma\left(\sum_{i=1}^{N}p_{(1,k)}^{t,i}\left\langle\mathbf{w}_{(1,k)},\mathbf{x}_i\right\rangle\right)\right]\\
&\leq\sum_{t=1}^{N}\mathbb{E}_{\epsilon}\left[\sup_{\|\mathbf{w}_{(1,k)}\|=M_1}\sum_{j=1}^{M}\epsilon_j p_{(2)}^{j,t}\sigma\left(\sum_{i=1}^{N}p_{(1,k)}^{t,i}\left\langle\mathbf{w}_{(1,k)},\mathbf{x}_i\right\rangle\right)\right]
\end{aligned}
\tag{22}
$$

For any fixed $t$, we have

$$
\mathbb{E}_{\epsilon}\left[\sup_{\|\mathbf{w}_{(1,k)}\|=M_1}\sum_{j=1}^{M}\epsilon_j p_{(2)}^{j,t}\sigma\left(\sum_{i=1}^{N}p_{(1,k)}^{t,i}\left\langle\mathbf{w}_{(1,k)},\mathbf{x}_i\right\rangle\right)\right]
$$

$$
\overset{(a)}{\leq}2\max_j\left|p_{(2)}^{j,t}\right|\cdot\mathbb{E}_{\epsilon}\left[\sup_{\|\mathbf{w}_{(1,k)}\|=M_1}\sum_{j=1}^{M}\epsilon_j\sigma\left(\sum_{i=1}^{N}p_{(1,k)}^{t,i}\left\langle\mathbf{w}_{(1,k)},\mathbf{x}_i\right\rangle\right)\right] \tag{23}
$$

$$
\overset{(b)}{\leq}2\,\mathbb{E}_{\epsilon}\left[\sup_{\|\mathbf{w}_{(1,k)}\|=M_1}\sum_{j=1}^{M}\epsilon_j\sigma\left(\sum_{i=1}^{N}p_{(1,k)}^{t,i}\left\langle\mathbf{w}_{(1,k)},\mathbf{x}_i\right\rangle\right)\right]
$$

The inequality (a) is due to the contraction property of ERC. The inequality (b) is due to the definition of graph attention matrix $\mathbf{P}$, i.e., the maximum value of entries in each row is equal to 1.

Put (19), (20), (22), (23) together, we get

$$
\mathbb{E}_{\epsilon}\left[\sup_{\|\mathbf{W}_{(1,k)}\|_F\leq M_1}\left\|\sum_{j=1}^{M}\epsilon_j\sum_{t=1}^{N}p_{(2)}^{j,t}\cdot\mathbf{z}_{(1)}^{t}\right\|_2\right]
$$

$$
\leq 4\sum_{k=1}^{K}\sum_{t=1}^{N}\mathbb{E}_{\epsilon}\left[\sup_{\|\mathbf{w}_{(1,k)}\|=M_1}\left(\sum_{j=1}^{M}\epsilon_j\sigma\left(\sum_{i=1}^{N}p_{(1,k)}^{t,i}\left\langle\mathbf{w}_{(1,k)},\mathbf{x}_i\right\rangle\right)\right)\right] \tag{24}
$$

Then we use the fact that $\sigma$ is $L_\sigma$ Lipschitz continuous and the contraction property of ERC. We further derive that

$$
4\sum_{k=1}^{K}\sum_{t=1}^{N}\mathbb{E}_{\epsilon}\left[\sup_{\|\mathbf{w}_{(1,k)}\|=M_1}\left(\sum_{j=1}^{M}\epsilon_j\sigma\left(\sum_{i=1}^{N}p_{(1,k)}^{t,i}\left\langle\mathbf{w}_{(1,k)},\mathbf{x}_i\right\rangle\right)\right)\right]
$$

$$
\overset{(a)}{\leq}4L_\sigma\sum_{k=1}^{K}\sum_{t=1}^{N}\mathbb{E}_{\epsilon}\left[\sup_{\|\mathbf{w}_{(1,k)}\|=M_1}\left\langle\sum_{j=1}^{M}\epsilon_j\sum_{i=1}^{N}p_{(1,k)}^{t,i}\mathbf{x}_i,\mathbf{w}_{(1,k)}\right\rangle\right]
$$

$$
\overset{(b)}{\leq}4L_\sigma\sum_{k=1}^{K}\sum_{t=1}^{N}\mathbb{E}_{\epsilon}\left[\sup_{\|\mathbf{w}_{(1,k)}\|=M_1}\left(\left\|\sum_{j=1}^{M}\epsilon_j\sum_{i=1}^{N}p_{(1,k)}^{t,i}\mathbf{x}_i\right\|_2^2\left\|\mathbf{w}_{(1,k)}\right\|_2^2\right)^{1/2}\right]
$$

$$
\leq 4L_\sigma M_1\sum_{k=1}^{K}\sum_{t=1}^{N}\left(\mathbb{E}_{\epsilon}\left[\left\|\sum_{j=1}^{M}\epsilon_j\sum_{i=1}^{N}p_{(1,k)}^{t,i}\mathbf{x}_i\right\|_2^2\right]\right)^{1/2} \tag{25}
$$

$$
\overset{(c)}{=}4L_\sigma M_1\sum_{k=1}^{K}\sum_{t=1}^{N}\left(\sum_{j=1}^{M}\left[\left\|\sum_{i=1}^{N}p_{(1,k)}^{t,i}\mathbf{x}_i\right\|_2^2\right]\right)^{1/2}
$$

$$
=4L_\sigma M_1\sum_{k=1}^{K}\sum_{t=1}^{N}\left(\sum_{j=1}^{M}\left[\left\|\mathbf{X}\mathbf{p}_{(1,k)}^{t}\right\|_2^2\right]\right)^{1/2},
$$

Where the inequality (a) is by Cauchy-Schwartz, and the inequality (b) is by Jensen's inequality. And the equality (c) follows the i.i.d. condition of Rademacher sequences with zero-mean.

Next, we continue to bound the rest,

$$4L_\sigma M_1 \sum_{k=1}^{K} \sum_{i=1}^{N} \left( \sum_{j=1}^{M} \left[ \left\| \mathbf{X} \mathbf{p}_{(1,k)}^t \right\|_2^2 \right] \right)^{1/2} \leq 4L_\sigma M_1 \sqrt{M} \sum_{k=1}^{K} \sum_{i=1}^{N} \left( \|\mathbf{X}\|_2 \left\| \mathbf{p}_{(1,k)}^t \right\|_2 \right)$$

$$\leq 4L_\sigma M_1 \sqrt{M} \sum_{k=1}^{K} \sum_{i=1}^{N} \left( \|\mathbf{X}\|_2 \left\| \mathbf{p}_{(1,k)}^t \right\|_1 \right) = 4L_\sigma M_1 \sqrt{M} \sum_{k=1}^{K} \sum_{i=1}^{N} \left( \|\mathbf{X}\|_2 \right)$$

$$\leq 4L_\sigma M_1 \sqrt{M} \sum_{k=1}^{K} \sum_{i=1}^{N} \left( \sup_{\|\mathbf{w}\|_2 = 1} \|\mathbf{X}\mathbf{w}\|_2 \right) \leq 4L_\sigma M_1 \sqrt{M} \sum_{k=1}^{K} \sum_{i=1}^{N} \left( \sup_{\|\mathbf{w}\|_2 = 1} \left[ \sum_{i=1}^{N} \left( \langle \mathbf{x}_i, \mathbf{w} \rangle \right)^2 \right]^{1/2} \right)$$

$$\leq 4L_\sigma M_1 \sqrt{M} \sum_{k=1}^{K} \sum_{i=1}^{N} \left( \sum_{i=1}^{N} \|\mathbf{x}_i\|_2^2 \right)^{1/2} \leq 4L_\sigma M_1 B K (N)^{3/2} M^{1/2},$$

where the first inequality is by $\|\mathbf{A}\mathbf{x}\|_2 \leq \|\mathbf{A}\|_2 \|\mathbf{x}\|_2$.

Combined with the earlier result, we get

$$\mathcal{R}(\mathcal{F}^c) \leq 4L_\sigma B K (N)^{3/2} M^{-1/2} M_1 M_2$$

## B  POOF OF THEOREM 2

We now turn to prove the Theorem 2. Our proof strategy will be the following. We first provide a classic theorem that was used to bound the expected loss based on the empirical loss and the upper bound of ERC of loss functions associated with GATs for the multi-time-step situation $\mathcal{F}$, then we derive this upper bound of ERC of loss functions of $\mathcal{F}$ by extending the upper bound of ERC of GATs with one-dimensional output for the single-time-step prediction $\mathcal{F}^c$, based on the existing theorem in literature.

*Proof.* From Theorem Mohri et al. (2018), we have the following holds for all $f$

$$\mathbf{E}(f) \leq \hat{\mathbf{E}}(f) + 2\mathcal{R}(g_\mathcal{F}) + 3\sqrt{\frac{\ln(2/\delta)}{2M}},$$

where $\mathbf{E}(f)$, $\mathcal{R}(g_\mathcal{F})$, and $\hat{\mathbf{E}}(f)$ are defined in section 2.4.

In order to extend to multi-dimensional vector-valued functions for MTSF, we will use the contraction inequality for the hypothesis class $\mathcal{F}$ of vector-valued functions $f \in \mathbb{R}^C$.

**Lemma 5** (Corollary 4 in Maurer (2016)). *let $F$ be a class of vector-valued functions $f = (f_1, \ldots, f_C) \in \mathbb{R}^C$, with each $f_c \in \mathcal{F}^c \subset \mathbb{R}^\mathcal{X}$, and let $\{\mathbf{x}_1, \ldots, \mathbf{x}_M\}$, $\{\mathbf{y}_1, \ldots, \mathbf{y}_M\}$ be a data set with each $\mathbf{x}_j \in \mathcal{X}$ and $\mathbf{y}_j \in \mathcal{Y}$. Let $\psi(\cdot, \cdot)$ be a 1-Lipschitz function mapping $\mathcal{V} \times \mathcal{Y}$ to $\mathbb{R}$, and associated to $\mathcal{F}$, where $\mathcal{F} \subset \mathcal{V}^\mathcal{X}$. Then we have*

$$\mathbb{E}_\epsilon \left[ \sup_{f \in \mathcal{F}} \sum_{j=1}^{M} \epsilon_j \psi((f(\mathbf{x}_j), \mathbf{y}_j) \right] \leq \sqrt{2} \, \mathbb{E}_\epsilon \left[ \sup_{f \in \mathcal{F}} \sum_{j=1}^{M} \sum_{c=1}^{C} \epsilon_{jc} f_c(\mathbf{x}_j) \right] \qquad (26)$$

*where $\epsilon_{cj}$ is the $c, j$-th entry of a $C \times M$ matrix of independent Rademacher variables.*

However, the RHS of equation 26 has supremum over all $f \in \mathcal{F}$ which is hard to compute and we can reduce it to scalar classes, and derive the following bound (Maurer, 2016):

$$\mathbb{E}_\epsilon \left[ \sup_{f \in \mathcal{F}} \sum_{j=1}^{M} \sum_{c=1}^{C} \epsilon_{jc} f_c(\mathbf{x}_j) \right] \leq \sum_{c=1}^{C} \mathbb{E}_\epsilon \left[ \sup_{f \in \mathcal{F}^c} \sum_{j=1}^{M} \epsilon_j f(\mathbf{x}_j) \right], \qquad (27)$$

Then we can derive the following upper bound for the loss function $g(f(\mathbf{x}), \mathbf{y})$ with $f(\mathbf{x})$ being vector-valued functions based on equation 26 and 27, where the RHS of equation 27 is related to $\mathcal{R}(\mathcal{F}^c)$:

$$\mathcal{R}(g_\mathcal{F}) \leq \sqrt{2} C \mathcal{R}(\mathcal{F}^c) \qquad (28)$$

$\square$

## C  PROOF THEOREM 3 AND THEOREM 4

*Proof.* For $l = L$, by the definition of network output and the Rademacher complexity, we have

$$
\mathcal{R}(\mathcal{F}^c) = \mathbb{E}_\epsilon \left[ \frac{1}{M} \sup_{f \in \mathcal{F}} \left| \sum_{j=1}^M \epsilon_j f(\mathbf{x}_j)^c \right| \right]
$$

$$
= \mathbb{E}_\epsilon \left[ \frac{1}{M} \sup_{\substack{f_{(L-1)} \in \mathcal{F}_{L-1} \\ \|\mathbf{w}_{(L)}^c\| \leq M_L}} \left| \sum_{j=1}^M \epsilon_j \sum_{t=1}^N p_{(L)}^{j,t} \cdot \left\langle \mathbf{z}_{(L-1)}^t, \mathbf{w}_{(L)}^c \right\rangle \right| \right]
$$

$$
\leq \frac{M_L}{M} \mathbb{E}_\epsilon \left[ \sup_{\substack{\|\mathbf{W}_{(L-1,k)}\|_F \leq M_{L-1} \\ f_{(L-2)} \in \mathcal{F}_{L-2}}} \left\| \sum_{j=1}^M \epsilon_j \sum_{t=1}^N p_{(L)}^{j,t} \cdot \mathbf{z}_{(L-1)}^t \right\| \right]
$$

so we get

$$
\frac{M}{M_2} \mathcal{R}(\mathcal{F}) \leq \mathbb{E}_\epsilon \left[ \sup_{\substack{\|\mathbf{W}_{(L-1,k)}\| \leq M_{L-1} \\ f_{(L-2)} \in \mathcal{F}_{L-2}}} \left\| \sum_{j=1}^M \epsilon_j \sum_{t=1}^N p_{(L)}^{j,t} \cdot \mathbf{z}_{(L-1)}^t \right\| \right]
$$

We denote the RHS as $\mathcal{R}(L)$.
We further unfold the expectation, we have

$$
\mathbb{E}_\epsilon \left[ \sup_{\substack{\|\mathbf{W}_{(L-1,k)}\|_F \leq M_{L-1} \\ f_{(L-2)} \in \mathcal{F}_{L-2}}} \left\| \sum_{j=1}^M \epsilon_j \sum_{t=1}^N p_{(L)}^{j,t} \cdot \mathbf{z}_{(L-1)}^t \right\| \right]
$$

$$
\leq 4 \sum_{k=1}^K \sum_{t=1}^N \mathbb{E}_\epsilon \left[ \sup_{\substack{\|\mathbf{w}_{(L-1,k)}\| = M_{L-1} \\ f_{(L-2)} \in \mathcal{F}_{L-2}}} \left( \sum_{j=1}^M \epsilon_j \sigma \left( \sum_{i=1}^N p_{(L-1,k)}^{t,i} \left\langle \mathbf{w}_{(L-1,k)}, \mathbf{z}_{L-2}^i \right\rangle \right) \right) \right]
$$

This follows the same reason with equation 24.
Now for any $l \in [L-1]$, we have

$$\mathbb{E}_\epsilon \left[ \sup_{\substack{\|\mathbf{w}_{(l,k)}\|=M_l \\ f_{(l-1)}\in\mathcal{F}_{l-1}}} \left( \sum_{j=1}^{M} \epsilon_j \sigma \left( \sum_{i=1}^{N} p_{(l,k)}^{t,i} \left\langle \mathbf{w}_{(l,k)}, \mathbf{z}_{l-1}^i \right\rangle \right) \right) \right]$$

$$\leq L_\sigma \, \mathbb{E}_\epsilon \left[ \sup_{\substack{\|\mathbf{w}_{(l,k)}\|=M_l \\ f_{(l-1)}\in\mathcal{F}_{l-1}}} \left\langle \sum_{j=1}^{M} \epsilon_j \sum_{i=1}^{N} p_{(l,k)}^{t,i} \mathbf{z}_{l-1}^i, \mathbf{w}_{(l,k)} \right\rangle \right]$$

$$\leq L_\sigma \, \mathbb{E}_\epsilon \left[ \sup_{\substack{\|\mathbf{w}_{(l,k)}\|=M_l \\ f_{(l-1)}\in\mathcal{F}_{l-1}}} \left\| \sum_{j=1}^{M} \epsilon_j \sum_{i=1}^{N} p_{(l,k)}^{t,i} \mathbf{z}_{l-1}^i \right\|_2 \|\mathbf{w}_{(l,k)}\|_2 \right]$$

$$= L_\sigma M_l \left( \mathbb{E}_\epsilon \left[ \sup_{\substack{\|\mathbf{W}_{(l-1,k)}\|_F \leq M_{l-1} \\ f_{(l-2)}\in\mathcal{F}_{l-2}}} \left\| \sum_{j=1}^{M} \epsilon_j \sum_{i=1}^{N} p_{(l,k)}^{t,i} \mathbf{z}_{l-1}^i \right\|_2 \right] \right)$$

Then we get the induction equation, which says

$$\mathbb{E}_\epsilon \left[ \sup_{\substack{\|\mathbf{W}_{(l,k)}\|_F \leq M_l \\ f_{(l-1)}\in\mathcal{F}_{l-1}}} \left\| \sum_{j=1}^{M} \epsilon_j \sum_{t=1}^{N} p_{(l+1,k)}^{j,t} \cdot \mathbf{z}_{(l)}^t \right\|_2 \right]$$

$$\leq 4 L_\sigma M_l \sum_{k=1}^{K} \sum_{t=1}^{N} \left( \mathbb{E}_\epsilon \left[ \sup_{\substack{\|\mathbf{W}_{(l-1,k)}\|_F \leq M_{l-1} \\ f_{(l-2)}\in\mathcal{F}_{l-2}}} \left\| \sum_{j=1}^{M} \epsilon_j \sum_{i=1}^{N} p_{(l,k)}^{t,i} \mathbf{z}_{l-1}^i \right\|_2 \right] \right)$$

$$\mathcal{R}(l+1) \leq 4 L_\sigma M_l \sum_{k=1}^{K} \sum_{t=1}^{N} \mathcal{R}(l)$$

By induction, we get

$$\mathcal{R}(l+1) \leq (4 L_\sigma K N)^{L-1} \prod_{l=1}^{L-1} M_l \mathcal{R}(1)$$

$$\leq (4 L_\sigma K N)^{L-1} \prod_{l=1}^{L-1} M_l \left( \mathbb{E}_\epsilon \left[ \left\| \sum_{j=1}^{M} \epsilon_j \sum_{i=1}^{N} p_{(1,k)}^{t,i} \mathbf{x}_i \right\|_2^2 \right] \right)^{1/2}$$

$$\leq (4 L_\sigma K N)^{L-1} \left( \prod_{l=1}^{L-1} M_l \right) B (NM)^{1/2}$$

Combining with the previous result, we get

$$\mathcal{R}(\mathcal{F}^c) \leq (4 L_\sigma K)^{L-1} \left( \prod_{l=1}^{L} M_l \right) B N^{L-1/2} M^{-1/2}$$

$\square$

## D  EXPERIMENT AND DATA

### D.1  EXPERIMENT SETUP

The model is trained by the Adam optimizer (Kingma & Ba, 2014). The learning rate is **1e-4**. The number of training epochs is **30**. The batch size is set to **5**. We split the dataset into three parts for training, validation and testing with a ratio of $0.6 : 0.2 : 0.2$. All the deep learning models, are implemented in Python with Pytorch and executed on a server with 8 NVIDIA GeForce GTX 2080Ti GPUs. The Nvidia rriver version is 470.141.03 and the CUDA version is 11.4.

Table 1: Experiment environment. The list only includes major packages. All the packages are installed using Anaconda and Pip.

| Package | Version |
| --- | --- |
| python | 3.9.13 |
| matplotlib | 3.5.2 |
| numpy | 1.23.1 |
| pandas | 1.4.4 |
| pytorch | 1.12.1 |
| torch-geometric | 2.1.0.post1 |
| torch-scatter | 2.0.9 |
| torch-sparse | 0.6.15 |

### D.2  NEURAL NETWORK ARCHITECTURE

We implement our neural network as a three-layer Graph Attention Neural network. This includes the input layer, one hidden layer, and the output layer. Each layer is a GATConv layer from the Pytorch-Geometric package [3]. We use ELU activation (Clevert et al., 2015) and Dropout (Srivastava et al., 2014) after both the input layer and the hidden layer.

For the number of heads variable, we change the number of attention heads of each layer and all layers use the same number of attention heads. For the number of neighbors variable, we adjust the dropout rate inside each layer's attention mechanism (different from the dropout layer) so that only a percentage of the nodes are considered when using the attention to aggregate information from a node's neighbors. For the weight norm variable, we adjust the bound of the Frobenius norm of the weight matrix inside each layer by using weight clipping. Each element of the weight matrix is clipped to the threshold to make sure the Frobenius norm of the matrix is less than or equal to the bound. For the number of layers variable, we adjust the number of hidden layers ranging from 1 to 8. For the input norm variable, we make sure each node's feature vector's L1 norm is less than or equal to a bound ranging from 1 to 28. For the number of labeled nodes variable, we use a ratio to adjust the size of the training dataset.

### D.3  MODEL HYPERPARAMETERS

Table 2: Default hyperparameters. When we study the impact of different values of a variable such as the number of heads, we keep all other variables fixed to the a set of same values.

| Hyperparameter | Value | Comment |
| --- | --- | --- |
| num-hid-layers | 1 | Number of hidden GATConv layers |
| num-heads | 2 | Number of attention heads; Same for all layers |
| num-neighbors | 0.1 | Neighbors to obtain attentions; Percentage of all nodes |
| num-labeled-nodes | None | Ratio of training set if not None |
| train-ratio | 0.6 | Ratio of training dataset |
| inputs-norm | 1.0 | Bound of the norm of the inputs |
| weights-bound | None | Bound of the norm of the model weights if not None |
| hidden-size | 32 | Out-channels of the input layer and hidden layer |
| lr | 1e-4 | Learning rate |
| dropout | 0.1 | Dropout rate after input layer and hidden layer |

---

[3] https://pytorch-geometric.readthedocs.io/en/latest/modules/nn.html

### D.4 DATA

We use the US daily stock market prices dataset from Kaggle [4].

### D.5 SOURCE CODE

The source code is shared via Dropbox [5]. Please refer to the included README file for detailed execution guide.

## E SUPPLEMENTAL EXPERIMENTS

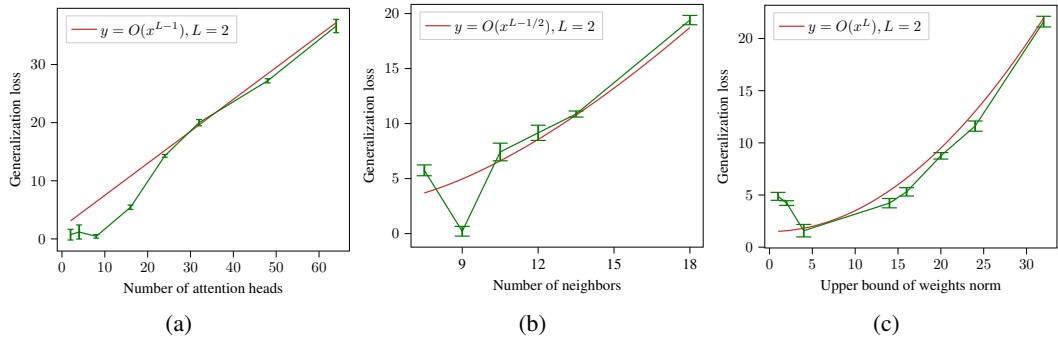

(a)             (b)             (c)

Figure 3: Additional experiment results using a two-layer GAT on three variables in the ERC whose generalization error is related to the number of layers. We run the experiment 20 times and obtain a standard deviation of the generalization error. (a) relationship between test loss and the number of attention heads. (b) relationship between test loss and the number of neighbors. (c) relationship between test loss and the upper bound of weight norm. The red line is a possible theoretical upper bound. The plots show that when L equals to 2, all test losses still generally conform to the big O of the theoretical upper bound.

In the experiment section, for all experiments, where the number of layers, $L$, needs to be fixed for studying the relationship of the generalization error with different GAT components that impact the ERC, including the number of attention heads, the number of neighbors, the upper bound of weights norm, we use a three-layer GAT to demonstrate Theorem 3. However, the upper bound of input norm and the number of labeled nodes affecting the ERC do not depend on the number of layers, thus, in our supplemental experiment for two-layer GATs, we only consider the three variables (the number of attention heads, the number of neighbors, and the upper bound of weights norm) that impact ERC and depend on the number of layers. Notably, Theorem 1 for generalization error bound of two-layer GATs is just a special case of Theorem 3 for deep GATs: when we let $L$ equal to 2 in Theorem 3, the generalization error bound is identical to the bound in Theorem 1 multiplied with some constants. We now give empirical results on two-layer GATs to see if it is consistent with Theorem 1 and Theorem 3.

In this appendix section, we include more experiment results from a two-layer GAT on three variables in the ERC. As Figure 3 shows, when the number of attention heads increases, the generalization error also increases at a linear rate after certain values. When the number of neighbors increases, the generalization error initially decreases when the number of neighbors is small, then starts increasing following the trend of a polynomial function. As for the upper bound of weight norm, the generalization error increases quadratically with the increment of the norm. The empirical results for all three variables generally conforms to the theoretical bound suggests by Theorem 1 as well as Theorem 3. For some inconsistency between the results and the reference red line or theoretical results at the the beginning, the reason, as we have discussed, can be due to the trade-off between approximation error and estimation error, since when the complexity of hypothesis class increases, the former decreases and the latter increases. In the long run, the estimation error or the ERC dominate the bound.

Thus, the results for $L = 2$ along with the three-layer experiments provide evidence for our theoretical findings (Theorem 1 and Theorem 3).

---

[4] https://www.kaggle.com/datasets/paultimothymooney/stock-market-data
[5] https://www.dropbox.com/sh/gpa3283fgpq0yx1/AAAYnzAmnIhVdp4KFXw5wTzAa?dl=0

# F SUPPLEMENT RESULTS

## F.1 MTSF UNDER FULLY-SUPERVISED SETTING INSTEAD OF SEMI-SUPERVISED SETTING

We consider an undirected graph $\mathcal{G} = (\mathcal{N}, \mathcal{E})$. $\mathcal{N} = (n_1, \ldots, n_N)$, $|\mathcal{N}| = N$, is a set of node labels representing the sources of $N$ time series. $\mathcal{E} \subset \mathcal{N} \times \mathcal{N}$ is the set of edges representing the connection between series. We let $\mathbf{x}_i \in \mathcal{X}$, $i \in [N]$ be a random variable representing the input feature vector of node $n_i$ for time series $i$. For node $i$, its random input feature $\mathbf{x}_i \in \mathcal{X} \subset \mathbb{R}^D$ is a multi-dimensional vector, which contains all the historical values from $T$ time steps, in other words, we let $\mathbf{x}_i = (\mathbf{x}_{i,t}, \ldots, \mathbf{x}_{i,t-T+1})$ be the concatenation of $T$ time steps; its true label $\mathbf{y}_i \in \mathcal{Y} \subset \mathbb{R}^C$ is the vector for the C-step-away values. We sample $n$ training data over $\mathcal{G}$, where $n = N \times V$ for some $V \in \mathbb{N}^+$. In other words, we have $V$ batches of samples over graph $\mathcal{G}$. For any risk function $g$ defined over $\mathcal{F}$, given the training set $S = \{(\mathbf{x}_1, \mathbf{y}_1), \ldots, (\mathbf{x}_n, \mathbf{y}_n)\}$ which includes $n$ samples from $\mathcal{X} \times \mathcal{Y}$ according to distribution $Q$, the expected/population risk $\mathbf{E}(f)$ and the empirical risk function $\hat{\mathbf{E}}(f)$ are defined as:

$$\mathbf{E}(f) = \mathbb{E}_{(\mathbf{x},\mathbf{y})\sim Q}[g(f(\mathbf{x}), \mathbf{y})], f \in \mathcal{F}. \quad (29) \qquad \hat{\mathbf{E}}(f) = \frac{1}{V}\frac{1}{\tilde{N}}\sum_{v=1}^{V}\sum_{j=1}^{\tilde{N}} g(f(\mathbf{x}_j), \mathbf{y}_j). \quad (30)$$

We introduce $1 \leq \tilde{N} \leq N$ as the *effective size* of data because the data among nodes corresponding to different time series are not independent. If $\tilde{N} = 1$, all of the time series are fully dependent. If $\tilde{N} = N$, they are mutually independent. So the $\tilde{N}$ characterizes the strength of the independence among different time series. Then we use $\tilde{n}$ as the *effective sample size* of data which are i.i.d, where $V \leq \tilde{n} \leq NV = n$. So the training set $S$ will contain $\tilde{n}$ data.

Given the inputs $\mathbf{X} = (\mathbf{x}_i, \ldots, \mathbf{x}_N)$ as multiple time series with each $\mathbf{x}_i$ as input feature for node $n_i$, the class of 2-layer GATs for MTSF $f$ maps $\mathbf{x}$ to the output $f(\mathbf{x})$ that represents a C-step-away prediction expressed in equation 2. We consider a subset of such class requiring each $f$ with a bounded weights norm, expressed as

$$\mathcal{F} = \left\{\mathbf{x} \mapsto f(\mathbf{x}) \in \mathbb{R}^C; \left\|\mathbf{W}_{(1,k)}\right\|_F \leq M_1, \left\|\mathbf{w}_{(2)}^c\right\| \leq M_2\right\}. \quad (31)$$

Furthermore, we also provide a model space $\mathcal{F}^c \subset \mathbb{R}^{\mathcal{X}}$ with a single dimensional output that corresponds to the $c$-th component of model output from $f(\mathbf{x})$ for the $c$-th time step, expressed as

$$\mathcal{F}^c = \left\{\mathbf{x} \mapsto f(\mathbf{x})^c, \left\|\mathbf{W}_{(1,k)}\right\|_F \leq M_1, \left\|\mathbf{w}_{(2)}^c\right\| \leq M_2, c \in [C]\right\}. \quad (32)$$

Here we first provide an upper bound of ERC of GAT class $\mathcal{F}^c$ for single dimensional output of MTSF.

**Theorem 6** (Upper Bound of ERC of GAT class $\mathcal{F}^c$ for MTSF). *Let the activation function $\sigma(\cdot)$ be $L_\sigma$-Lipschitz continuous, and also satisfy $\sigma(0) = 0$ and $\sigma(\alpha z) = \alpha\sigma(z)$ for all $\alpha \geq 0$. Assume that the $L_2$-norm of the feature vector $\mathbf{x}$ comes from a bounded domain $\mathcal{X} = \{\mathbf{x} : \|\mathbf{x}\| \leq B\}$. Assume that the Frobenius norm of every weights matrix in the first layer of the GAT class is bounded, namely, $\left\|\mathbf{W}_{(1,k)}\right\|_F \leq M_1$ with some constant $M_1 > 0$ for every $k$. Also, the norm of the weights vector of the second layer of the GATs is bounded, $\left\|\mathbf{w}_{(2)}^c\right\| \leq M_2$, where $c \in [C]$, with some constant $M_2 > 0$. Let $\mathcal{N}_i$ denote the neighborhood of node $i$ (including $i$), let the number of neighbors of each node be identical, namely, for some common constant $N_e \in \mathbb{N}^+$, assume $N_e := |\mathcal{N}_i|$ for all node $i \in \mathcal{N}$, furthermore, we consider the most general formulation, which allows every node to attend on every other node, i.e., $N_e = N$.*

*Then let $\mathcal{R}(\mathcal{F}^c)$ be the ERC defined in the definition 7 for GAT class $\mathcal{F}^c$ in the definition 32, given the $\tilde{n}$ sized input set $\{\mathbf{x}_1, \ldots, \mathbf{x}_{\tilde{n}}\}$, then we have*

$$\mathcal{R}(\mathcal{F}^c) = \mathcal{O}(L_\sigma B K M_1 M_2 (N_e)^{3/2} \tilde{n}^{-1/2}).$$

Similar proof details can be found in §A.

**Theorem 7.** *Define the hypothesis class $\mathcal{F}$ as the definition 15. We suppose $g$ is Lipschitz with constant $L_g$. Then for any $\delta \in (0, 1)$, with probability at least $1 - \delta$, for all $f \in \mathcal{F}$, we have*

$$\mathbf{E}(f) \leq \hat{\mathbf{E}}(f) + 2\sqrt{2}CL_g\mathcal{R}(\mathcal{F}^c) + 3\sqrt{\frac{\ln(2/\delta)}{2\tilde{n}}},$$

*where we have* $\mathcal{R}(\mathcal{F}^c) = \mathcal{O}(L_\sigma BKM_1M_2(N_e)^{3/2}\tilde{n}^{-1/2})$ *from Theorem 6.*

## F.2 Results under Supervised Setting

In this section, we provide the results of experiments under the fully-supervised setting. The multivariate time series have about 1500 stocks, and all of these stocks are used for training and testing.

We ensure that the training and testing data do not have any overlap. The results for the relationship between the upper bound of the generalization error of the model and variables in the ERC are shown in the Figure 4.

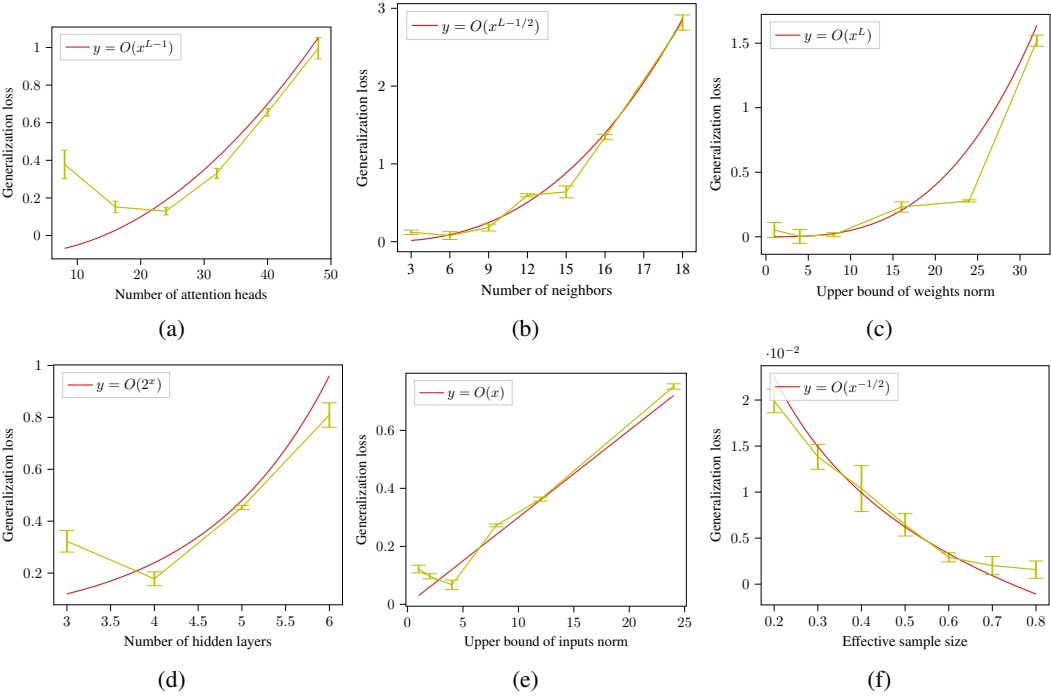

Figure 4: Experiment results on six variables in the ERC. We run the experiment 20 times and obtain a standard deviation of the generalization error. (a) relationship between test loss and the number of attention heads. (b) relationship between test loss and the number of neighbors. (c) relationship between test loss and the upper bound of weight norm. (d) relationship between test loss and the number of (hidden) layers. (e) relationship between test loss and the upper bound of input norm. (f) relationship between test loss and the training data set size. The red line is a possible theoretical upper bound. The plots show that all test losses generally conform to the big O of the theoretical upper bound.

From Figure 4 we can see that the overall pattern for different variables are still consistent with those in Figure 1. For the last variable, we use the size of the training data set instead of the number of the labeled nodes. Our Theorem 6 reports that the ERC has a polynomial $\mathcal{O}(\tilde{n}^{-1/2})$'s dependence on the *effective sample size*. As we can see from Figure 4f, the generalization loss decreases when the training data set size increases. The red curve based on the theoretical value matches the pattern of the yellow curve based on our experiments, though slightly lower than the yellow curve after certain values of training data set size. This is because we need more samples in the experiment to satisfy the effective samples in terms of the generalization error.

Table 3: Comparison of Weight-bounded GAT with Baselines ASTGCN, GDN

|  | Metrics ($10^{-3}$) | | | |
|  | MSE | MAE | STD(MSE) | STD(MAE) |
| --- | --- | --- | --- | --- |
| Weight-bounded GAT | **2.04** | **30.61** | **0.9** | **10.55** |
| ASTGCN Guo et al. (2019) | 7.14 | 36.77 | 6.61 | 17.41 |
| GDN Deng & Hooi (2021) | 8.11 | 46.31 | 3.62 | 24.21 |

### F.3 COMPARISON WITH OTHER BASELINES

So far, two current works (Guo et al., 2019; Deng & Hooi, 2021) use GAT-based models to model multiple time series data, showing better performance in accuracy over other traditional linear methods (e.g., VAR), neural-network based methods (e.g., LSTM), and graph-network based methods (e.g., GNNs and GCNs). Both works differ from ours as they do not consider any control of model variables (e.g., the weight matrix norm) and they lack theoretical guarantees in terms of the generalization error. GDN is proposed by Deng & Hooi (2021) and is used for anomaly detection in multivariate time series. ASTGCN is developed by Guo et al. (2019) for multiple traffic flow forecasting. We use GDN and ASTGCN as baselines to compare with Weight-bounded GAT. To adapt them into our settings, we modify the output layer of GDN to perform the regression instead of classification task. We also keep other hyper-parameters the same across three methods. Table 3 reports their average test error and standard deviation from 20 runs. We use two evaluation metrics, the MSE and MAE. Additionally, we also use figure 5 to show their generalization loss difference.

As Table 3 and Figure 5 show, the Weight-bounded GAT performs better than the GDN and ASTGCN. Apart from using graph attention mechanism, GDN also learns a directed graph to model the causal-effect relationship between different nodes, which can capture the asymmetric dependency patterns. Learning a graph structure can better dynamically capture the relationship between nodes, but also increase the learning time and model complexity. In the meantime, while using attention between nodes, ASTGCN also considers the attention along the time axis. Our method does not consider attention along the time as the features in every time step has already embedded the historical information. Compared to GDN and ASTGCN, the weight-bounded control in our method plays an important role in controlling the generalization error, which is also verified in our theoretical analysis. This result further corroborated the outperformance of our method over the GAT-based SOTA methods for MTSF.

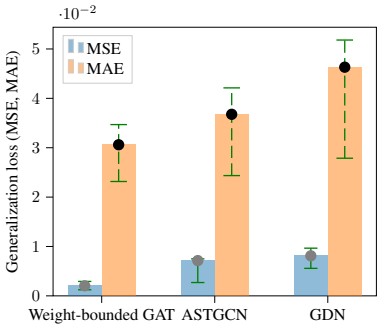

Figure 5: Test loss of three methods. The train-test pipeline runs 20 times over 20 random seeds. The Weight-bounded GAT has a better test loss than the GDN and ASTGCN regarding the first quantile, third quantile and the mean of the test loss.

### F.4 MODEL ARCHITECTURE

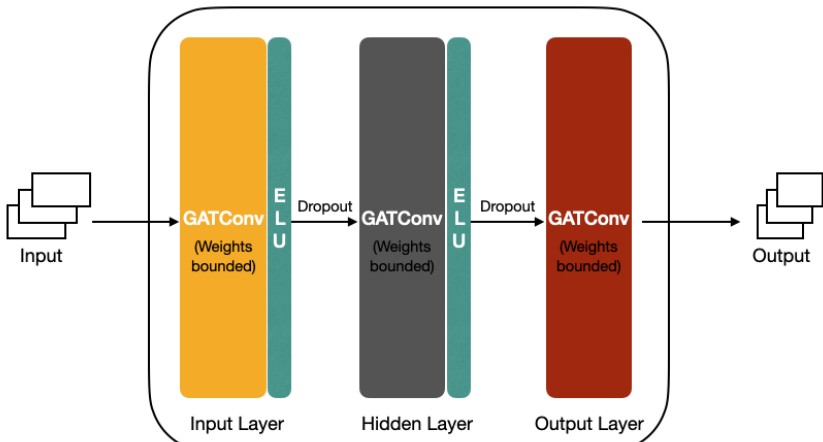

Figure 6: The demonstration of the three-layer Weight-bounded GAT model we use. The input layer and the hidden layer are both followed by an ELU activation and a dropout. Each GAT layer is implemented using a Pytorch-Geometric GATConv layer.

