# OpenReview forum: "Multivariate Time Series Forecasting By Graph Attention Networks With Theoretical Guarantees"
_ICLR.cc/2023/Conference — Submitted to ICLR 2023_

### Official Review · Reviewer_7dJx · 2022-10-22

**Confidence:** 2
**Correctness:** 3
**Technical Novelty And Significance:** 2
**Empirical Novelty And Significance:** Not applicable
**Recommendation:** 5

**Clarity, Quality, Novelty And Reproducibility:**

Clarity and Quality --Good:
The paper is well-written easy to follow and the notations are clean and consistent. I looked into the proof and followed the math and I believe they are correct (however I might be mistaken).

Novelty -- Significant:
Most work on multivariate time series is purely empirical, this paper proposes a graph attention network with theoretical guarantees with respect to generalization error bounds. The paper shows that this can be done by bounding the weight matrix.

Reproducibility -- Good:
The paper provided the code to replicate the experiments in section 6 showing that the theoretical bounds match the empirical error generated by the model.

**Details Of Ethics Concerns:**

One very small note:  The authors have certified that the URL could not be used to find the author's identity.  However, if I download the code in the supplementary material it directs me to a google drive that has a user name. Personally, for me, it makes no difference whatsoever however I think for future submissions I believe the authors should take more care to hide their identity.

**Strength And Weaknesses:**

Strength:
The paper provides theoretical guarantees on graph attention by bounding weight matrix for multivariate time series forecasting.
The paper is clearly written and easy to follow.
The paper shows empirically that test losses conform to the big O of the theoretical upper bound.

Weakness:

The title of the paper is "MULTIVARIATE TIME SERIES FORECASTING BY GRAPH ATTENTION NETWORKS WITH THEORETICAL GUARANTEES" and the area chosen by the authors where the paper falls is application. So one would expect to see the performance of the proposed method on multivariate time series and compare the performance to other methods. Yet the paper has no empirical experiments showing the performance of forecasting tasks. This makes me question the capabilities of the proposed GAT when compared to other neural methods for practical forecasting applications.

**Summary Of The Paper:**

The paper proposes a norm-bounded graph attention network (GAT) for multivariate time series forecasting. The norm of the weight matrix in the proposed model is bounded to ensure the generalization error bound, the paper proves this theoretically. The experimental section in the paper shows that the generalization performance depends on the number of attention heads, the number of neighbors at each given node, the upper bound of weights norm, the number of hidden layers, the upper bound of inputs norm, and the number of labeled nodes. Their empirical results match the theoretical expectations (this is shown in figure 1).


**Summary Of The Review:**

The main weakness of the paper is that it has no empirical experiments comparing the performance of GAT to other networks. While I appreciate the theoretical guarantees multivariate time series forecasting is a very practical topic and convincing the AI community to use a method must come with some strong empirical results. Proving error bounds is great but it is not enough, unfortunately. If the authors would like to provide this paper as a purely theoretical paper the title and area should have been different so more theoretical reviewers could have been assigned.
If the paper had strong empirical results I would have happily accepted the paper.

---

> ### Author Response · Authors · 2022-11-14
> **Response to Reviewer 7djx**
>
> We thank Reviewer 7dJx for the detailed and insightful feedback. We are encouraged by the positive comments that our paper is theoretically sound, well-written and easy to follow. We appreciate the reviewer's time to corroborate the correctness of our proof. We have revised our manuscript accordingly and added more experiments as suggested by reviewer 7dJx. Below is our point-to-point response to the reviewer's concerns.
>
> > Q1: The title of the paper is "MULTIVARIATE TIME SERIES FORECASTING BY GRAPH ATTENTION NETWORKS WITH THEORETICAL GUARANTEES" and the area chosen by the authors where the paper falls is application. So one would expect to see the performance of the proposed method on multivariate time series and compare the performance to other methods. Yet the paper has no empirical experiments showing the performance of forecasting tasks. This makes me question the capabilities of the proposed GAT when compared to other neural methods for practical forecasting applications.
>
> We thank the reviewer for the helpful suggestions. We have followed these suggestions and provided a baseline experiment in the appendix of the revised paper.
>
> - As mentioned in our paper, the graph-network-based methods are preferred over traditional linear time series analysis methods and neural networks without graph because traditional methods fail to consider the variable inter-dependency in the multivariate time series. The most recent attention-based method utilizes the weight importance of neighbors to build representation and has shown better performance than GNN/GCN in benchmark classification tasks [3]. We identified two recent studies by [1] [2] in the literature that have used attention-based methods to make the multivariate time series forecasting. Their results outperform methods that did not incorporate attention. Therefore, to evaluate our proposed method (the weight-bounded GAT), we add an additional experiment by comparing our methods with these two attention-based methods. Based on the prediction accuracy, our weight-bounded GAT outperforms theirs with accuracy. We provide the result in the revised paper（Appendix F.3）and attach the code as well.
>
> - We also want to use this rebuttal discussion to clarify our goal. As we mentioned, the numeric and experimental successes of GATs for MTSF notwithstanding, theoretical understandings of the underlying mechanisms of GATs for MTSF still need to be improved. Therefore, our first goal is to provide theoretical guarantees for the MTSF by GATs. Along this direction, we found that the generalization performance could be improved by controlling the weight matrix norm, the most dominant variable in our theoretical bound. Thus, we propose a general method (idea) to control the weight matrix Frobenius norm and verify it theoretically and empirically. We show that a careful GAT structure design could improve the model’s generalization performance for MTSF, and our paper provides a guideline for such a design. Practically, our method can help improve training without mindless parameter tuning. We believe this information is helpful for researchers in the application area and thus we put our paper into the application area.
>
>
> > Q2: The main weakness of the paper is that it has no empirical experiments comparing the performance of GAT to other networks. While I appreciate the theoretical guarantees multivariate time series forecasting is a very practical topic and convincing the AI community to use a method must come with some strong empirical results. Proving error bounds is great but it is not enough, unfortunately. If the authors would like to provide this paper as a purely theoretical paper the title and area should have been different so more theoretical reviewers could have been assigned. If the paper had strong empirical results, I would have happily accepted the paper.
>
>
>
> -  We have added more experiments to address the major concern over the lack of empirical results. Particularly, we use two attention-based methods as the baseline and compare them with our methods. The results clearly support the outperformance of our method. We also believe that the error bound suggested in our work can help application researchers design better GAT models. More details can be found in our above response to Q1.
>
> ---
> We greatly appreciate Reviewer 7dJx for the positive comments and constructive suggestions. Any change in the review score that reflects your agreement or recognition of our response will be appreciated. Please let us know if there are other questions or concerns. Thank you!

---

> > ### Comment · Reviewer_7dJx · 2022-11-24
> > **Thank you**
> >
> > I want to thank the authors for their response and for the extra experiments in Appendix F3.
> > However, a single dataset comparing two other models is not enough, I understand that the rebuttal time is very limited and it is not possible to conduct extensive experiments. So I think the authors should take more time working on the empirical section of the paper and resubmit.

---

> > > ### Author Response · Authors · 2022-11-24
> > > **Response to Reviewer 7dJx's concern that only two other methods have been compared, and more data are needed.**
> > >
> > > We thank reviewer 7dJx again. We would like to provide a prompt response to reviewer 7dJx's concerns. To our best knowledge, only two methods based on attention with graph networks are used for multivariate time series forecasting in the literature. Both methods outperform conventional methods not based on attention. Therefore, it should be sufficient only to compare our method with these two attention based methods.  We will appreciate it if reviewer sends us more references for graph attention network based methods for MTSF.  Data access involves several concerns including a suitable structure for MTSF and privacy preservation. We will attempt to locate more data for our method. This will be our focus in the recent period and can be completed before the final presentation.

---

### Official Review · Reviewer_yXLj · 2022-10-24

**Confidence:** 3
**Correctness:** 4
**Technical Novelty And Significance:** 2
**Empirical Novelty And Significance:** 2
**Recommendation:** 5

**Clarity, Quality, Novelty And Reproducibility:**

Clarity and quality are good. The paper is nicely written and easy to follow. The experiments are carefully designed to demonstrate the tightness and implications of theoretical claims. The subject studied in the paper is novel, although not very surprising.

**Details Of Ethics Concerns:**

The code is uploaded to a Google Drive where I can see the name of the owner. I have no idea about the relationship between the owner and the authors. It makes no difference for me but I'd like to raise it.

**Strength And Weaknesses:**

The paper attempts to provide a theoretical foundation to an important problem: the use of graph attention networks for multivariate time series forecasting. Using graphical information to capture correlations among variables in multivariate time series forecasting has gained a lot of interests recently. Although various GNN-based methods have been proposed and are shown to deliver state-of-the-art empirical performance, there is few rigorous study on the generalization error of these methods. Therefore the subject studied in this paper is very interesting. In addition, the paper is nicely written. The overall presentation is clear and easy to follow. The authors also carry out empirical experiments to validate all theoretical claims.

However, on the other side, I have a few concerns/questions regarding the practical relevance of the theoretical setting in this work to multivariate time series forecasting.

- Typically in multivariate time series forecasting, we do not have access to future C-step-away values of any series. This comes from the fact that in forecasting we never know what is going to happen in the future. However, in this paper, the authors assume that we are given the true future C-step-away values for a subset of time series. This assumption seems to be counter-intuitive and not very realistic. Can the authors comment on the assumption?

- The authors assume that each node in the graph has the same degree. Moreover, in Theorem 1 the authors claim that "we consider the most general formulation, which allows every node to attend on every other node". I think that requiring each node to have the same degree, e.g., requiring that every node to attend to every other node, is in fact a limitation rather than being general. The general setting should be without assuming anything about the structure of the graph, or at least without assuming that the graph is degree-regular or complete. I think the authors should make it clear in the text about the limitations imposed by the assumptions.

- In general, I feel like the setting in this paper, i.e, problem formulation in section 2.1 and the graph structure in section 2.2, is not really the typical setting for time series forecasting. It feels more like graph-based semi-supervised regression. For example, I couldn't find any characteristics in the problem setting that are special to time series forecasting. Moreover, I think that, in general, semi-supervised setting in more typical for classification rater than forecasting. Please correct me if I am wrong.

- Because of the above points, I think that there is a gap between what the paper claims to solve and what the paper actually solves. It might be more appropriate if the paper has been framed as providing generalization bound for graph attention network on semi-supervised regression tasks.

Finally, a minor question about the experiments: How did you construct the graph for the stock pricing dataset when you vary the node degrees? Is it kNN graph?

**Summary Of The Paper:**

This work proves generalization error bounds of multivariate time series forecasting by graph attention network, under the assumption that every node in the graph has the same degree. The theoretical analyses are based on upper bounding empirical Rademacher complexity of the class of functions represented by the graph attention network. The derived generalization bounds depend on the norm of weight matrices, the number of neighbors, the number of attention heads, and the number of layers. The theoretical claims are further validated by empirical results.

**Summary Of The Review:**

The paper provides a solid theoretical analysis on the generalization error of graph attention network for multivariate time series forecasting. However, I have some concerns about the relevance of the problem setting in this paper to multivariate time series forecasting.

---

> ### Author Response · Authors · 2022-11-14
> **To Reviewer yXLj: Response to the question about the construction of the graph.**
>
> > Question: "How did you construct the graph for the stock pricing dataset when you vary the node degrees? Is it kNN graph?"
>
> As our above answer to concern 2 mentioned, the graph is not manually constructed. It is learned by the model using the attention mechanism and sparsified by the top K control.
>
> ---
> We greatly appreciate Reviewer yXLj for the positive comments and constructive questions. We have addressed all the points raised by the reviewer. Please let us know if there are other questions or concerns. Thank you!

---

> ### Author Response · Authors · 2022-11-14
> **Response to reviewer yXLj: Clarification of a concern about the paper's achievement.**
>
> > Concern 4: "Because of the above points, I think that there is a gap between what the paper claims to solve and what the paper actually solves. It might be more appropriate if the paper has been framed as providing generalization bound for graph attention network on semi-supervised regression tasks."
>
> We also attach more experiment results that include the forecasting performance in a classic supervised setting and the comparison between our model and the SOTA GAT-based multivariate time series forecasting model (Appendix F.1, F.2, F.3). We hope these and the above explanation are enough to clarify that there is no gap between what the paper claims to solve and what the paper actually solves.

---

> ### Author Response · Authors · 2022-11-14
> **To Reviewer yXLj: Clarification of a concern about the setting of the paper.**
>
> > Concern 3: "In general, I feel like the setting in this paper, i.e, problem formulation in section 2.1 and the graph structure in section 2.2, is not really the typical setting for time series forecasting. It feels more like graph-based semi-supervised regression. For example, I couldn't find any characteristics in the problem setting that are special to time series forecasting. Moreover, I think that, in general, semi-supervised setting in more typical for classification rather than forecasting. Please correct me if I am wrong."
>
> As we clarify above, the setting in the paper is actually not the classic semi-supervised setting. It is more of a tweak of the supervised setting, where in the training phase, we use the supervised loss and ensure that the training time steps do not overlap with the testing time period, which is also a classic time series forecasting setting. We call it semi-supervised in the paper only because that we allow some of the nodes' responses to be missing in the training phase and calculate the supervised loss using a subgraph. Consequently, it is not simply a graph-based semi-supervised regression. Additionally, both the graph structure and the node attributes are specific to time series forecasting in the sense that each node represents a time series and we use a graph-based model to capture the inter-correlations between each variate in the multivariate time series forecasting task. We are not the first to adopt a graph-based setting in time series forecasting. There are several other works with a similar setting [1][2].
>
> We really appreciate the reviewer for raising the question on the semi-supervised setting, which makes us realize that there is confusion in our terminology. We hope the clarification above can clear the confusion. Additionally, we also attach more experiment results in a very typical supervised setting where the only difference with the original setting is that the entire graph is used in the training phase.
>
> - [1] Shengnan Guo, Youfang Lin, Ning Feng, Chao Song, and Huaiyu Wan. Attention based spatial-
> temporal graph convolutional networks for traffic flow forecasting. In Proceedings of the AAAI conference on artificial intelligence, volume 33, pp. 922–929, 2019
> - [2] Ailin Deng and Bryan Hooi. Graph neural network-based anomaly detection in multivariate time series. In Proceedings of the AAAI Conference on Artificial Intelligence, volume 35, pp. 4027–4035,2021.

---

> ### Author Response · Authors · 2022-11-14
> **To Reviewer yXLj: Clarification of a concern about each node in the graph has the same degree.**
>
> > Concern 2: "The authors assume that each node in the graph has the same degree. ... I think that requiring each node to have the same degree, e.g., requiring that every node to attend to every other node, is in fact a limitation rather than being general. The general setting should be without assuming anything about the structure of the graph..."
>
> We do not assume that each node in the real graph has the same degree. This impression may arise from both the method we use to process the graph and the simplification of the theory model. It has been shown in the literature that the message passing mechnism in graph neural network does not work well when the number of the neighbors of a node is too large, which causes the over-smoothing issue [1]. The intuition of this over-smoothing issue is that those unimportant neighbors' information dilates important neighbors' information so that they increase the noise ratio of the target node and degrade the final performance. Thus, a sparse graph is preferred and the method we use to ensure a sparse graph is to select only the top k neighbors in the message passing based on the correlation ranking. Thus, after taking the top K neighbors, every node has the same degree.
>
> For the concern related to the generality, the reviewer thinks that we should not assume any graph structure, which is the same as our opinion in the paper. As explained in the above paragraph, the top K control naturally makes degree of all nodes the same. This top K control only sparsifies the graph. When the value of K is set to the maximum that is equal to the total number of nodes, we allow every node to attend to other nodes in the graph. It is noteworthy that in this case, the degree to which a node is attended is completely dependent on the attention scores learned by the model. When the attention of a specific node is very close to 0, its information almost does not pass on in the message passing mechanism, equivalent to no edge connected to this node, informing the generality of our work. In comparison, if the K is set to some constants smaller than the total number of nodes, then the node has no freedom to attend to certain nodes in the graph. Furthermore, as we can see, there is no specific graph structure assumption being made in our work. The graph structure is decided by the attention scores learned by the model.
>
> [1] D. Chen, Y. Lin, W. Li, P. Li, J. Zhou, and X. Sun, “Measuring and relieving the over-smoothing problem for graph neural networks from the topological view,” in AAAI, 2020.

---

> ### Author Response · Authors · 2022-11-14
> **To Reviewer yXLj: Clarification of a concern about the use of future C-step-away values.**
>
> We thank Reviewer yXLj for the detailed and insightful feedback. We are happy that Reviewer yXLj thinks our paper is easy to follow and has a good quality.
>
> We have updated our manuscript and clarified the claims, as suggested by Reviewer yXLj. Below we address Reviewer yXLj's concerns in detail.
>
> > Concern 1: "However, in this paper, the authors assume that we are given the true future C-step-away values for a subset of time series. This assumption seems to be counter-intuitive and not very realistic. Can the authors comment on the assumption?"
>
> We do not have any assumption regarding access to future C-step-away values of any series in the testing phase. In our experiments, future values are not included in the testing phase. In the training phase, under both semi-supervised and fully supervised settings, it is necessary and correct to have the true response because we need to calculate a supervised loss. This so-called "C-step-away future data" in the training phase is actually part of the history data. For example, suppose that the current time is $t$, the $C$-step-away future time is $t+1,...,t+C$, the variable that we want to predict is $Y$ and the inputs to the forecasting model is $X$. Then, in the training phase of the model $f$, under any time $t$, the forward pass is $Loss(f(X_{t-7, t-6, ..., t}), Y_{t+1,...,t+C})$. In the text, we call $Y_{t+1,...,t+C}$ the $C$-step-away future data, which might cause confusions. Actually, it is just future data relative to $X_{t-7, t-6, ..., t}$. In the testing phase, we make sure that there is no overlap between the training data time period and testing data time period, and we use $f(X_{t+C+1, ..., t+C+7})$ to predict the target variable $Y_{t+C+7+1, ..., t+C+7+C}$. The naming may cause confusions. It is also possible that some of the confusions arise from the semi-supervised settings where only some of the nodes in the graph are used to calculate the loss in the training phase, which is different from assuming awareness of future values of some series in the testing phase. This semi-supervised setting is closer to the assumption that some nodes'  responses (the stock price in this case) are missing in the historical training data. So, the assumption underlying the semi-supervised setting is actually more general in the sense that we allow some nodes' responses to be missing in the training phase. As for the temporal dimension, we strictly ensure that all the time points in the training phase precede the earliest time point in the testing phase. Thus, there is no data leakage problem.
>
> Thank you so much for raising this concern and giving us the opportunity to further clarify it.

---

### Official Review · Reviewer_Hkz8 · 2022-10-25

**Confidence:** 3
**Correctness:** 2
**Technical Novelty And Significance:** 2
**Empirical Novelty And Significance:** 3
**Recommendation:** 3

**Clarity, Quality, Novelty And Reproducibility:**

Clarity seems to have much room for improvement. One of the issues might be that the paper does not clearly explain the forecasting framework itself. The problem setting does not seem to be a pure forecasting as some of the y values are visible. But it is not very clear to me. If theoretical analysis of the GAT framework (rather than forecasting), that's fine per se. However, in that case, the authors should clearly mention the scope of the paper upfront. The current introduction sounds as if they were boasting better forecasting accuracies. This may be a bit confusing. The apparent lack of clarity significantly affects the reproducibility of the paper as well.

**Strength And Weaknesses:**

Strength

- Novel incorporation of GAT into a multi-variate time-series forecasting framework.

- Conducted a theoretical analysis of the GAT parameter under the Frobenius-norm-based loss.


Weakness

- Despite that the authors claim that incorporation of GNN/GCN will improve the prediction performance, the authors do not provide a clear-cut description of the forecasting model itself.

- Similarly, it seems the authors did not evaluate the forecasting performance if I understand correctly. This gives the impression that the first half and the second half are disconnected and do not deliver a coherent message.



**Summary Of The Paper:**

This paper presents a multivariate time-series forecasting framework that combines GAT with a forecasting framework. The paper provides detail about the GAT framework. The way they integrate it seems to have been inspired by Wu et al. IJCAI 19 but differs in the choice of the specific GAT parameterization. The second half is about their theoretical analysis based on their empirical risk minimization framework.

**Summary Of The Review:**

This might be a potentially solid work from a theoretical perspective, but the overall impression is that it was submitted prematurely before careful proof reading.

---

> ### Author Response · Authors · 2022-11-14
> **To reviewer Hkz8: Response to concerns about paper's introduction, reproducibility, and maturity.**
>
> > Q6: The current introduction sounds as if they were boasting better forecasting accuracies. This may be a bit confusing.
>
> - Our current introduction mentioned two contributions. Firstly, we develop a weight-bounded GAT method for MTSF. In the rebuttal period, we added more experiments to compare our method to the state-of-the-art attention-based methods for MTSF (Appendix F.3). We show that our method has better forecasting accuracy.
>
> - Secondly and more importantly, we for the first time developed a theoretical guarantee to bound the generalization error with the GATs for MTSF. We provided insights into the relationship between the architecture of the GAT model and its generalization performance. We also empirically verified our theoretical findings by conducting large-scale complex stock price prediction experiments.
>
>
> > Q7: The apparent lack of clarity significantly affects the reproducibility of the paper as well.
>
> - We provide the experiment setup and data in the appendix D. We share the training codes and provide the readme file to run the experiment.
>
>
>
>
> > Q8: This might be a potentially solid work from a theoretical perspective, but the overall impression is that it was submitted prematurely before careful proof reading.
>
>
> - We thank the reviewer for making this suggestion. We have double-checked the manuscript carefully and addressed the problems as much as possible by using the fully-supervised setting, adding the baselines to evaluate our proposed method, and providing the framework in the appendix.
>
>
> ---
> We greatly appreciate Reviewer Hkz8's positive comments and constructive suggestions. We have updated our paper to address all the points raised by the reviewer. Please let us know if there are other questions or concerns. Thank you!

---

> ### Author Response · Authors · 2022-11-14
> **To reviewer Hkz8: Clarification of concern about the forecasting setting.**
>
> > Q5: The problem setting does not seem to be a pure forecasting as some of the y values are visible. But it is not very clear to me. If theoretical analysis of the GAT framework (rather than forecasting), that's fine per se. However, in that case, the authors should clearly mention the scope of the paper upfront.
>
> - We do not have any assumption regarding access to future y values of any series in the testing phase. In our experiments, future values are not included in the testing phase. In the training phase, under both semi-supervised and fully supervised settings, it is necessary and correct to have the true response because we need to calculate a supervised loss. This so-called "C-step-away future y" in training phase is actually part of the history data. For example, suppose that the current time is $t$, the $C$-step-away future time is $t+1,...,t+C$, the variable that we want to predict is $Y$ and the inputs to the forecasting model is $X$. Then, in the training phase of the model $f$, under any time $t$, the forward pass is $Loss(f(X_{t-7, t-6, ..., t}), Y_{t+1,...,t+C})$. In the text, we call $Y_{t+1,...,t+C}$ the $C$-step-away future y, which might cause confusions. Actually, it is just future data relative to $X_{t-7, t-6, ..., t}$. In the testing phase, we make sure that there is no overlap between the training data time period and testing data time period, and we use $f(X_{t+C+1, ..., t+C+7})$ to predict the target variable $Y_{t+C+7+1, ..., t+C+7+C}$. The naming may cause confusions.
>
> - It is also possible that some of the confusions arise from the semi-supervised settings where only some of the nodes in the graph are used to calculate loss in the training phase, which is different from assuming awareness of future values of some series in the testing phase. We set some label values missing in the training phase because the source paper of GATs [3] evaluated its performance on the semisupervised problems. To theoretically provide its generalization guarantee without too much variation, we also consider semi-supervised forecasting.
>
> - Furthermore, this semi-supervised setting is closer to the assumption that some nodes'  responses (the stock price in this case) are missing in the historical training data. So, the assumption underlying the semi-supervised setting is actually more general in the sense that we allow some nodes' responses to be missing in the training phase. More importantly, we claim that the choice of semi- or fully supervised settings does not affect our conclusion and contribution to the literature.
>
> - Methods in this paper can also be applied to a fully-supervised setting and our theory is still valid with a slight adjustment to the bound: our theoretical bound on the Rademacher complexity will be adjusted
> $O((4L_{\sigma}K)^{L-1}\prod_{l=1}^{L}M_l(B N^{L-1/2}n^{-1/2}))$ where $n$ is the effective sample size. We also experiment in the fully-supervised setting, and we put the results into our revised paper (Appendix F.2). The performance still surpasses prior methods.
>
> - We realize this is an important concern since other reviewers have also raised it, so we have added the theory and experiments for the fully-supervised setting (Appendix F.1, F.2, F.3). Thanks for the reviewer's question.
>
> ---
> [3] Petar Veliˇckovi ́c, Guillem Cucurull, Arantxa Casanova, Adriana Romero, Pietro Lio, and Yoshua Bengio. Graph attention networks. arXiv preprint arXiv:1710.10903, 2017.

---

> ### Author Response · Authors · 2022-11-14
> **To reviewer Hkz8: Response to the question about the lack of explanation of the forecasting framework .**
>
> > Q4: One of the issues might be that the paper does not clearly explain the forecasting framework itself.
>
> - We describe the current forecasting framework in section 2. Section 2.1 is about the time series forecasting problem. Section 2.2 is about the graph attention network model. In section D of the appendix, we also have the detailed setup of the method, including the neural network architecture, experiment parameter, and data preparation. We did not add a framework figure due to space limitations in the initial submission because our theoretical analysis concentrates on the generalization error for the GAT-based method under the multivariate time series setting. We thought interested readers could refer to the vanilla GATs paper [3] for more details.
>
> - We thank the reviewer's suggestion and add the forecasting framework in appendix F.4.

---

> ### Author Response · Authors · 2022-11-14
> **To reviewer Hkz8: Response to the question about no evaluation about the forecasting performance.**
>
> > Q3: Similarly, it seems the authors did not evaluate the forecasting performance if I understand correctly. This gives the impression that the first half and the second half are disconnected and do not deliver a coherent message.
>
> - We thank the reviewer for bringing up this point. We want to use this rebuttal to make our claims more clear. Currently, in Figure 1, we report the relationship between test error and six crucial variables in our model. In Figure 2, we show that the proposed Weight-bounded GAT outperforms the vanilla GAT significantly in the test error. Furthermore, we report the MSE in our results and provide the experiment parameters in the appendix. Finally, figure 2 shows that a careful GAT structure design could improve the model’s generalization performance for MTSF, and our paper offers a guideline for such a design.
>
> - To make our claims more clear and better present our method, in the rebuttal period, we also add the baseline experiments for the state-of-the-art GAT-based method on MTSF (Appendix Section F.3). The baseline is based on the recent works done by [1] and [2]. We compared their prediction test error with our weight-bounded GAT. The results show that our method outperforms the SOTA methods. We provide the result in the appendix and attach the code as well.
>
> - Practically, our method can facilitate training without mindless parameter tuning. Suppose in a job for tuning the number of attention heads (K), if we find the test error increases when K changes from 3 to 4, then there is no need to further increase it to 5. As Figure 1 shows, test error begins to increase quadratically when the number of heads exceeds certain values.
>
> ---
> [1] Shengnan Guo, Youfang Lin, Ning Feng, Chao Song, and Huaiyu Wan. Attention based spatial-
> temporal graph convolutional networks for traffic flow forecasting. In Proceedings of the AAAI conference on artificial intelligence, volume 33, pp. 922–929, 2019
>
> [2] Ailin Deng and Bryan Hooi. Graph neural network-based anomaly detection in multivariate time series. In Proceedings of the AAAI Conference on Artificial Intelligence, volume 35, pp. 4027–4035,2021.
>
> [3] Petar Veliˇckovi ́c, Guillem Cucurull, Arantxa Casanova, Adriana Romero, Pietro Lio, and Yoshua Bengio. Graph attention networks. arXiv preprint arXiv:1710.10903, 2017.

---

> ### Author Response · Authors · 2022-11-14
> **To reviewer Hkz8: Response to the question about no forecasting model for GNN/GCN.**
>
> > Q2: Despite that the authors claim that incorporation of GNN/GCN will improve the prediction performance, the authors do not provide a clear-cut description of the forecasting model itself..
>
> - We describe our forecasting framework in section 2. Section 2.1 is about the time series forecasting problem. Section 2.2 is about the graph attention network model. In section D of the appendix, we also have the detailed setup of the method, including the neural network architecture, experiment parameter, and data preparation. We did not add a framework figure due to space limit in the initial submission because our theoretical analysis concentrates on the generalization error for the GAT-based method under the multivariate time series setting. We thought interested readers could refer to the vanilla GATs paper [3] for more details.
>
> - We thank the reviewer's suggestion and add the forecasting framework in the appendix F.4.
>
> - GNN/GCN is not a subject of our paper. We reviewed several previous work about GNN and GCN because we want to give a complete story about the history of MTSF. As mentioned in the introduction, although GNN/GCN outperforms prior methods in MTSF tasks, [1] [2] showed that attention-based models even demonstrate better performance over GNN/GCN.

---

> ### Author Response · Authors · 2022-11-14
> **To reviewer Hkz8: Response to the concern about the similarity of our paper with the Wu et al. IJCAI 19 work.**
>
> We thank Reviewer Hkz8 for the detailed and helpful feedback. We are encouraged that our paper is regarded potentially solid from a theoretical perspective by Reviewer Hkz8.
>
> Below we address questions and concerns raised by Reviewer Hkz8 in detail.
>
> > Q1: The way they integrate it seems to have been inspired by Wu et al. IJCAI 19 but differs in the choice of the specific GAT parameterization.
>
> - We thank the reviewer for providing us with this reference, although we had a chance to read it after the submission. However, we consider our work differs from theirs significantly in the following aspects:
>
>    - (i). The goal of the paper by Wu et al. is twofold: 1) handle divergent data types using a heterogeneous graph, and 2) improve the representation of user profiling with limited label data using GATs, which is completely unrelated to multivariate time series forecasting (MTSF), the major focus of our paper.
>    - (ii). Their main task is multi-class classification, while ours is regression. Moreover, their task does not involve any time series data. By contrast, we deal with multi-variate time series data, which contains interdependency among different series. One of our goals is to use the graph-based models to capture this interdependency.
>    - (iii). Wu et al. use a heterogeneous graph that is split into divergent subgraphs, with each subgraph as a separate neural network layer, and then they build the attention between subgraphs. We consider a single graph with topology based on neighbors.
>    - (iv). Their attention methods are inconsistent in the sense that three different attention methods originated from [3], [4], and [5] are used in their work. Here we call [3] GAT, [4] Vanilla, and [5] Laplacian. In their so-called "User Profiling" task, they use Vanilla for the attribute-to-item subgraphs, the Laplacian or GAT for the user-to-user subgraphs, and Vanilla for the item-to-user subgraphs. The arbitrary usage of three different methods adds to complxity and can be, sometimes, unnecessary, as GAT [3] has already showed better performance than Vanilla [4] and Laplacian [5]. Furthermore, they do not provide any reasoning or give any intuition about their choice. By contrast, we only consider one attention method, GAT, but with different controls (weight norm, input norm, number of attention heads, number of neighbors, and number of layers), aiming to understand its mechanism in terms of generalization performance.
>    - (v). Wu et al. integrate three previous methods into one framework. We propose a new method of weight-bounded GAT based on our theory and verify its performance theoretically and empirically. Most importantly, their work lacks theoretical guarantees on the generalization error of the method, although the work gives a specific application scenario.
>
>
> - We aim to study the attention-based method for MTSF, especially the GATs. As mentioned in our paper,  recent studies have shown that attention-based methods have the potentials to solve the inter-dependency between multiple variables in MTSF. Moreover, the performance these attention-based methods is empirically better than those without considering attention. However, the numeric and experimental successes of GATs for MTSF notwithstanding, theoretical understandings of the underlying mechanisms of GATs for MTSF are still limited. Therefore, our first goal is to provide theoretical guarantees for the MTSF by GATs. Along this direction, we found that the generalization performance could be improved by controlling the weight matrix norm. Thus, we propose a general method (idea) to control the weight matrix Frobenius norm and verify it theoretically and empirically. Building on these theoretical findings, our paper's second goal is to introduce the weight-bounded GAT to improve the attention-based method performance for MTSF.
>
> - We rigorously focus on advancing theoretical understanding and developing methods, rather than integrating the GATs with MTSF. Our work is not a reparametrization of this prior work by Wu et al., as their work did not take the weight norm and input norm into consideration.
>
> ---
> [3] Petar Veliˇckovi ́c, Guillem Cucurull, Arantxa Casanova, Adriana Romero, Pietro Lio, and Yoshua Bengio. Graph attention networks. arXiv preprint arXiv:1710.10903, 2017.
>
> [4] Xiangnan He, Zhankui He, Jingkuan Song,
> Zhenguang Liu, Yu-Gang Jiang, and Tat-Seng Chua. NAIS: Neural attentive item similarity model for recommendation. IEEE Transactions on Knowledge and Data Engineering, 30(12):2354–2366, 2018.
>
> [5] Thomas N. Kipf and Max Welling. Semi supervised classification with graph convolutional networks. In ICLR, 2017

---

### Author Response · Authors · 2022-11-14
**General Response: Summary of Paper Updates**

We thank all reviewers for their recognition of the novelty and contribution of our work. All their comments are helpful and have been reflected in the updated paper.

Here we briefly outline the updates to the revised submission based on the reviews. Then, we provide point-to-point response in our replies to each reviewer.

### Paper Updates

We create a new section F in the Appendix, where reviewers can easily locate all the updates and changes. In the later stage, if needed, we can integrate these changes into our main texts.

- **[Appendix F.1, F.2]**: (1) We included the fully supervised setting for multivariate time series forecasting (MTSF). Under this setting, we introduced the associated empirical risk minimization setup, the new theoretical bound, and the corresponding experiments (Hkz8, yXLj).
- **[Appendix F.3]**: We added more baselines for our method from SOTA graph attention network-based methods for MTSF [1] [2]. We included new experiments using the fully-supervised settings to compare the baselines with our proposed "weight-bounded GAT" method in terms of prediction accuracy. (Hkz8, 7dJx)
- **[Appendix F.4]**: We clarified our model framework description by including a demonstration graph of the model framework. (Hkz8)
- **[The entire paper]**: We polished the writing and clarified the description of claims. We fixed several typos and duplicate references. We proofread the paper thoroughly to correct all grammar errors.

We greatly appreciate all reviewers' suggestions. Furthermore, we hope that our updates and responses have addressed reviewers' questions and concerns. Please let us know if there are further questions.

---
- [1] Shengnan Guo, Youfang Lin, Ning Feng, Chao Song, and Huaiyu Wan. Attention based spatial-
temporal graph convolutional networks for traffic flow forecasting. In Proceedings of the AAAI conference on artificial intelligence, volume 33, pp. 922–929, 2019

- [2] Ailin Deng and Bryan Hooi. Graph neural network-based anomaly detection in multivariate time series. In Proceedings of the AAAI Conference on Artificial Intelligence, volume 35, pp. 4027–4035,2021.

---

### Decision · Program_Chairs · 2023-01-20

**Decision:**

Reject

**Justification For Why Not Higher Score:**

see above

**Justification For Why Not Lower Score:**

see above

**Metareview: Summary, Strengths And Weaknesses:**

Unfortunately, the reviewers were not enthusiastic enough about this paper for it to be considered for acceptance at ICLR 2023. There are just too many papers that reviewers were much more enthusiastic about this year, and ICLR has a very low acceptance rate. The authors are encouraged to take the reviewer comments very seriously, even if there are things you disagree with, and make sure that all issues are addressed, and any potential sources of confusion are completely eliminated, in the next version of the paper. Also, please ensure that these are addressed in the initial paper submission for that next conference.


**Summary Of Ac-Reviewer Meeting:**

see above